# Increasing extreme precipitation variability plays a key role in future record-shattering event probability
Iris de Vries [1] ✉, Sebastian Sippel [2], Joel Zeder [1], Erich Fischer [1] & Reto Knutti [1]

Climate events that break records by large margins are a threat to society and ecosystems. Climate change is expected to increase the probability of such events, but quantifying these probabilities is challenging due to natural variability and limited data availability, especially for observations and very rare extremes. Here we estimate the probability of precipitation events that shatter records by a margin of at least one pre-industrial standard deviation. Using large ensemble climate simulations and extreme value theory, we determine empirical and analytical record shattering probabilities and find they are in high agreement. We show that, particularly in high emission scenarios, models project much higher record-shattering precipitation probabilities in a changing relative to a stationary climate by the end of the century for almost all the global land, with the strongest increases in vulnerable regions in the tropics. We demonstrate that increasing variability is an essential driver of near-term increases in record-shattering precipitation probability, and present a framework that quantifies the influence of combined trends in mean and variability on record-shattering behaviour in extreme precipitation. Probability estimates of record-shattering precipitation events in a warming world are crucial to inform risk assessment and adaptation policies.

In recent years weather and climate extremes have become the order of the day. Not only record-high temperatures but also destructive heavy precipitation events hit many world regions more and more frequently and with unprecedented intensity[1,2]. Floods are the second most costly natural disaster worldwide (after storms), hence, with increasing extreme precipitation frequency and intensity, potential economic damage due to flooding will also increase[3,4]. Increasing society's resilience to the changing character of weather and climate extremes demands adaptation of critical infrastructure and emergency plans. A shortcoming of infrastructure design and crisis management is its reliance on current risks, such as maximum possible intensities of weather conditions based on statistics of historical observations[5,6]. Additionally, the increasing degree to which humanity shapes the natural environment reduces the vividness of disaster memory and may introduce a sense of false security[7]. If events of alleged impossible intensities then do occur, severe damage and losses often result e.g. refs. 6,8,9. An essential task thus lies in improving resilience to the changing risks of a future climate. This requires a complex, collaborative approach involving many stakeholders, integrating technical, organisational, economic and social aspects[5,10,11]. In this study, we address an essential input to this process, namely, estimates of

probabilities and intensities of unseen precipitation events in a changing climate.

In addition to the long-term effects of anthropogenic climate change, natural climate variability strongly shapes regional trends in climate variables on shorter (decadal to multidecadal) timescales[12]. Accordingly, natural variability can lead to decades without local trends or broken records, despite climate change[13]. Against such a backdrop, the chance that the climate catches up to the long-term trend with a sudden jump is considerable. We evaluate the probability of such jumps, which we refer to as record shattering. In this work, record-shattering events are defined as events that break, indeed shatter, the historical maximum value (record) by a margin of at least one pre-industrial standard deviation, defined as the mean standard deviation of the period 1850–1949.

In a stationary climate the theoretical probability that the latest observation of any independent and identically distributed random variable —for example, annual maxima of temperature or precipitation—breaks the record (i.e. exceeds the maximum of the observed values thus far) is a simple function of time: For the $t$th observation, the record-breaking probability is $1/t$[14]. If the variable's distribution does not change (stationary), the record-breaking probability thus asymptotically approaches zero with time.

[1]Institute for Atmospheric and Climate Science, ETH Zürich, Zürich, Switzerland. [2]Leipzig Institute for Meteorology, Leipzig University, Leipzig, Germany. ✉ e-mail: iris.devries@env.ethz.ch

For record shattering the probability decreases faster with time and the $1/t$-relationship is modified by the distributional characteristics of the variable in question. Yet, as for record breaking, the record-shattering probability in a stationary climate asymptotically approaches zero with time.

Record-breaking and -shattering probabilities in our changing climate deviate from this course[15]. It has been shown that record-shattering heat becomes more likely with increasing warming trends[16], exemplified by events such as the 2021 Pacific Northwest heat wave where local temperature records were broken by more than 5 °C[17,18], equivalent to a record jump of about one standard deviation and anomalies of more than four standard deviations[19]. For extreme heat, the local warming rate of maximum temperature—and not the warming level—drives record-shattering probability[16].

Record-shattering precipitation events have not yet been investigated. We hypothesise that these will also occur more frequently with climate change as the water holding capacity of the atmosphere increases approximately exponentially with a rate of about 7% per degree of warming[20–23]. Albeit this does not directly translate to the change in extreme precipitation with warming—precipitation efficiency and dynamical processes modify the scaling rate[24]—, exponential scaling of extreme precipitation with local temperature has been shown to hold in models and observations, especially in mid- to high latitudes[22,23]. Additionally, increases in precipitation intensity scale with event rareness, which is particularly relevant for record-shattering precipitation since it implies that the most extreme precipitation events—ones that have the potential to shatter records—see a relatively stronger intensification than more moderate ones[25–27]. Yet, the high temporal and spatial variability of heavy precipitation is expected to influence precipitation record shattering characteristics and complicate the response. The large temporal fluctuations in precipitation might increase the likelihood of jumps, and changing atmospheric stability and circulation patterns intensify precipitation increases in some regions while suppressing them in others e.g. refs. 24,28,29. The exponential, skewed extreme precipitation intensification with warming, in combination with its high spatial and temporal variability, raises the question of how record-shattering precipitation probabilities will evolve in a changing climate.

In this study, we show that simulated future record-shattering probabilities for seasonal maximum precipitation increase in response to climate change on both global and local scale, using data from the Community Earth System Model 2 Large Ensemble (CESM2-LE)[30], supported by several models from the Coupled Model Intercomparison Project phase 6 (CMIP6)[31], forced with historical forcing followed by shared socio-economic pathway (SSP) 3–7.0[32], a relatively high warming scenario. We separate the contributions of trends in the mean and trends in variability using extreme value statistics and analytical relationships, leading to a conceptual framework that describes the near-term and long-term evolution of precipitation record-shattering probabilities as a function of distributional changes. The effects of climate change on record-shattering probability depend on the relative changes in mean and variability as well as on the climatological distributional characteristics of seasonal regional extreme precipitation, leading to complex behaviour. Additional complexity is introduced by scenario uncertainty, model uncertainty and error: precipitation record-breaking and -shattering behaviour differs between climate models and between simulations and the real world due to, among other things, uncertainty in future emissions, climate sensitivity, and forced dynamical changes, small scale processes that are not resolved in models, and natural variability. While acknowledging these uncertainties and the limitations of climate model simulations, we identify a framework for the analysis of changing record-shattering precipitation probability. Our main finding is that positive trends in variability lead to near-term increases in record-shattering probability, whereas steepening trends in the mean drive long-term increases in record-shattering probability.

## Results and discussion
### Simulated record-shattering events resemble 2021 European floods

In the summer of 2021, parts of Germany, Luxembourg, Belgium and The Netherlands witnessed a flooding event heavier than ever observed in the region, due to heavy rainfall lasting several days. Record-shattering rainfall was reported at several observational stations[33–35]. The black timeseries in Fig. 1a shows the observed anomaly (E-OBS[36]) of the annual maximum of daily precipitation (Rx1d) for extended boreal summer (May–September, MJJAS) averaged over a representative region (red box in Fig. 1b–i), with a pink marker indicating the event. This event is heavier than any other observation by a large margin, and a generalised extreme value (GEV) distribution fitted to the region-averaged data in the 71 years preceding, but not including, the event shows that the event exceeded the 1000-year return level. Note that this is based on a simple stationary GEV-analysis of the local observations alone, for an extensive analysis of return levels, see e.g. refs. 33,34.

Do climate models simulate similarly extreme events, and if so, do they look physically plausible? To justify investigation of record-shattering precipitation in climate models, we compare the observed event with an event simulated by CESM2-LE[30]. The dashed black line in Fig. 1a shows the regionally averaged Rx1d timeseries of the single CESM2-LE member featuring the heaviest simulated event in the future period (red marker), which is our event of interest for physical comparison. The 1000-year return level of the GEV distribution fitted to this member in the same way as to the observed data is exceeded by this event, indicating that CESM2-LE is indeed capable of simulating extremely rare events that exceed very high return levels, similar to the observed 2021 event. Additionally, the simulated Rx1d anomalies lie within a range comparable to that of the observed Rx1d anomalies, shown in Supplementary Fig. S1.

In Fig. 1b–e and f-i we compare the meteorological conditions of the observed 2021 event to those of the simulated event, and see a high degree of similarity. Note that the simulated event was selected based on it being the most extreme future event, not based on resemblance to the observed event. We see high updraft anomalies collocated with the precipitation maximum, which is consistent with the expected convective nature of heavy summer rain. Vertically integrated water vapour transport (IVT) shows a strong moisture flux coming from the North Atlantic and North Sea that is diverted into a cyclonic circulation around the surface low (negative sea level pressure (SLP) anomaly) above the region of interest, picking up moisture from the Mediterranean on its downstream flank. The magnitudes of the variables shown cannot easily be compared since the two events are of different intensity and, even more so, since the events take place in a different climatological background and the data sources (observations and simulations) differ considerably. The qualitative character of the drivers and the within-dataset anomalies, however, are convincingly similar. Supplementary Fig. S2 shows a second simulated event with comparable physical characteristics, reducing the likelihood that this similarity is a coincidence. The congruence between the simulated and observed event strengthens confidence in the physical consistency of the simulated very extreme precipitation events in CESM2-LE and justifies further investigation of the statistical properties of such events using the simulated data. Importantly, this congruence does not imply that the statistical characteristics of precipitation as simulated in CESM2-LE are comparable to observations in general: there are large model uncertainties surrounding extreme precipitation statistics and projections, and also observational uncertainty is large due to sparse coverage and short timeseries. In the following, our results thus pertain to the large ensemble climate model context in which we conduct this study. Additional work is needed to connect these findings to our real climate.

### Global record-shattering precipitation probability increases relative to stationary climate

In the following, we present a statistical analysis of record-shattering precipitation in the 100-member ensemble of CESM2-LE under historical

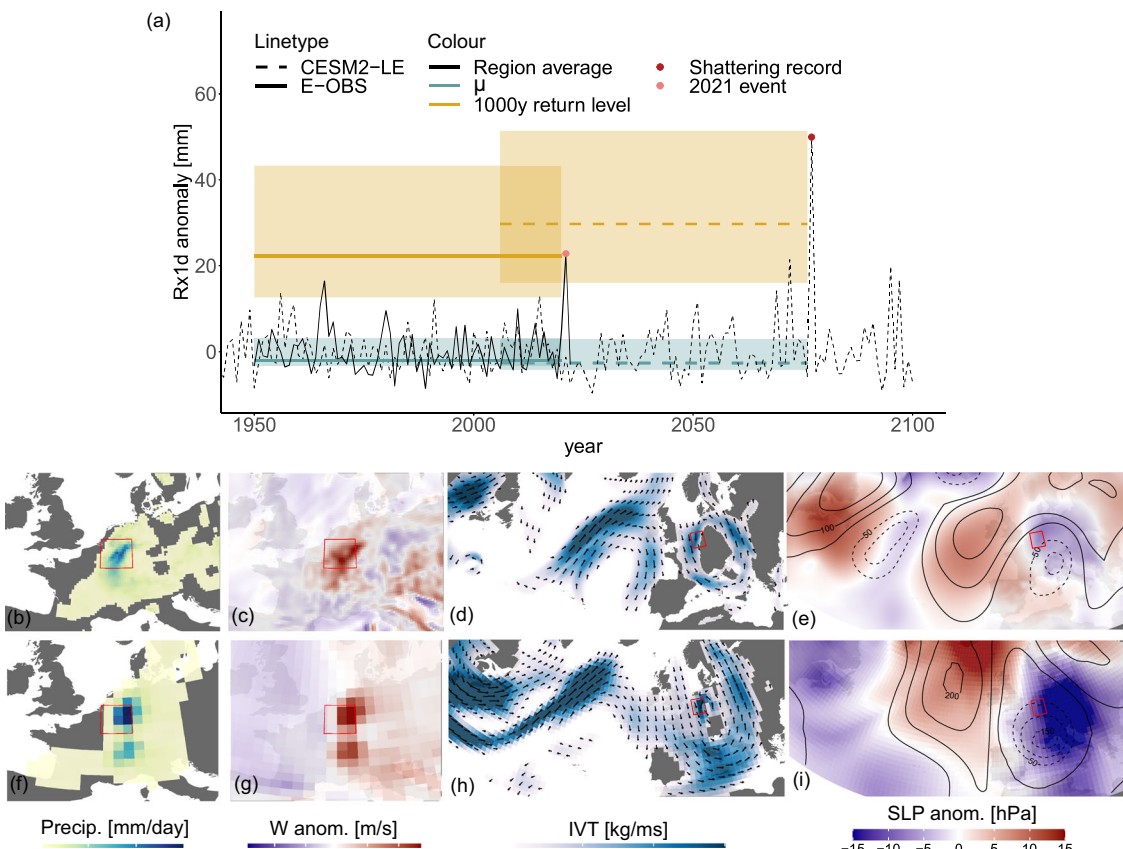

**Fig. 1 | Temporal evolution of extreme precipitation and meteorological situation associated with record-shattering precipitation events in climate model simulations and observations. a** MJJAS region mean Rx1d anomalies w.r.t. 1950–2022 for the region hit by floods in July 2021[33] (red box) from E-OBS (solid) and from the CESM2-LE ensemble member featuring the most extreme event in the region (dashed). Generalised extreme value (GEV) statistics are shown for both datasets, the following quantities are all based on a GEV distribution fitted to the 71 years of region-mean data preceding the event: yellow shows the 1000-year return level, shading its 95% confidence intervals; blue shows the location parameter $\mu$; blue shading shows the GEV distribution's interquartile range. **b** E-OBS, and (**c–e**) ERA5 data for relevant meteorological variables for the 2021 precipitation event day: precipitation (Precip.), vertical updraft anomaly (W anom.), vertically integrated water vapour transport (IVT), geopotential height anomaly at the 500 hPa level (Z500 anom.), and sea level pressure anomaly (SLP anom.), E-OBS and ERA5 anomalies are computed relative to the 1950–2022 average (dataset length). **f–i** show the same variables for the CESM2-LE most extreme event in the region, CESM2-LE anomalies are relative to the 1850–2100 average (dataset length). See 'Meteorological event analysis' for more details on the data and methods. Maps in all figures in this paper are based on the maps R-package[73].

forcing followed by SSP3-7.0. We focus on 5-year block maxima of precipitation—5yRx1d—, to ensure we only include rare, truly extreme events. Record shattering is path dependent, and is thus evaluated in each individual member, at each grid cell. The 5yRx1d event of member $i$ at time $t$ is considered record-shattering if it exceeds the maximum 5yRx1d value in ensemble member $i$ since $t = 1$, i.e. the year 1850, by a margin larger than the record-shattering threshold $c_{rs}$. This means that a given value can constitute a record-shattering event in one ensemble member but not in another, due to the different history of the members. The record-shattering threshold we use is the ensemble standard deviation of 5yRx1d averaged over the historical reference period 1850–1949. In short, if expression (1) is true, 5yRx1d$_{i,t}$ is considered a record-shattering event.

$$5yRx1d_{i,t} > \max(5yRx1d_{i,1}, ..., 5yRx1d_{i,t-1}) + c_{rs} \quad (1)$$

We assess the spatial pattern of precipitation record-shattering probability changes by means of 5yRx1d record-shattering probability ratios. These ratios are defined for each 5-year period as the 5yRx1d record-shattering probability in SSP3-7.0 $P_{CC}$ divided by the 5yRx1d record-shattering probability in a counterfactual stationary climate $P_{ref}$. $P_{CC}$ is determined in two ways: $P_{CCe}$ (empirical) is based on the 5yRx1d data directly and constitutes the fraction of ensemble members that shatter their own historical record, and $P_{CCa}$ (analytical) is the analytical record-

shattering probability derived from GEV distributions fitted to the 5yRx1d data under climate change, as explained below. $P_{ref}$ is always estimated analytically and based on GEV distributions fitted to the 5yRx1d data from 1850–1949, which we assume to reflect pre-industrial 5yRx1d.

$$P(x_t > \max(x_1, ..., x_{t-1}) + c_{rs}) = \int_{-\infty}^{\infty} f_t(x + c_{rs}) \prod_{i=1}^{t-1} F_i(x) \, dx \quad (2)$$

The analytical record-shattering probabilities $P_{CCa}$ and $P_{ref}$ at time $t$ are computed using Eq. (2), in which $x$ denotes 5yRx1d, and $c_{rs}$ represents the record-shattering threshold. For $P_{CCa}$, the probability density function $f(x)$ and cumulative distribution function $F(x)$ are time dependent and defined by grid cell specific non-stationary GEV distributions fitted to the 5yRx1d data. Three parameters characterise these GEV distributions: location parameter $\mu$, scale parameter $\sigma$, and shape parameter $\xi$. Non-stationarity in our application implies that $\mu$ and $\sigma$ vary with time as linear functions of their covariates; we here use the smoothed 5yRx1d ensemble mean as covariate for $\mu$, and the smoothed 5yRx1d ensemble standard deviation as covariate for $\sigma$. These covariates were chosen so as to reflect the response of the 5yRx1d mean and variability to external greenhouse gas forcing in the corresponding GEV parameters. The grid cell specific trends in $\mu$ and $\sigma$ can thus be considered measures of the local forced response in mean and

variability of 5yRx1d. Note that $\mu$ and $\sigma$ are not equivalent to the mean and the standard deviation, but they are qualitatively comparable. $\xi$ is kept constant with time for reasons of accuracy and tractability. For $P_{ref}$, $f_i(x)$ and $F_i(x)$ are time-invariant and based on stationary GEV distributions fitted to 5yRx1d data from 1850–1949. Additional details and GEV validation tests can be found in 'Extreme value analysis' and Supplementary Section S2.

We thus obtain empirical and analytical probability ratios, defined as $\frac{P_{CCe}}{P_{ref}}$ and $\frac{P_{CCa}}{P_{ref}}$. In Fig. 2a–d we show the 2070–2099 average probability ratios, reflecting how many times more likely it is that a 5yRx1d record is shattered in any 5-year window in 2070–2099 in the SSP3-7.0 scenario compared to a world without climate change. Figure 2a and b show empirical probability ratios, and Fig. 2c and d show analytical probability ratios for the extended hemispheric seasons November-March (NDJFM) and May–September (MJJAS).

Figure 2 a and b demonstrate that the CESM2-LE simulations with climate change according to SSP3-7.0 feature more frequent 5yRx1d record shattering relative to a world without climate change almost everywhere over land in both seasons, reflected by the area where empirical 5yRx1d probability ratios exceed 1. This is noteworthy, given that not all these locations see a pronounced positive mean 5yRx1d trend (see Supplementary Fig. S11), suggesting that distributional changes beyond a mean trend are of import for record-shattering behaviour. The few regions where probability ratios are smaller than 1 and climate change leads to lower record-shattering probabilities compared to a stationary climate (purple), correspond to regions with negative mean trends in 5yRx1d (see Supplementary Fig. S11), which have been linked to dynamical processes[24].

The probability ratios exhibit a physically intelligible pattern, with the highest values in low-latitude tropical and monsoon regions, where climatological levels and relative trends in mean and variability of 5yRx1d are high (see Supplementary Fig. S11). Previous studies found similar patterns featuring high sensitivity of precipitation extremes to warming in low latitudes, both in models and observations[29]. Effects of dynamical changes on the deep convective systems that drive extreme precipitation events in those regions are likely to play a role[24,37,38]. Also mountain ranges, where both thermodynamic and dynamic effects are at play[29], stand out. This is concerning from a flood-risk perspective, especially in combination with the recent finding that snow-to-rain transition amplifies the increase in rainfall in elevated regions[39].

We also distinguish signatures of seasonality in the patterns—the most evident being the more northern latitudinal position of the intertropical convergence zone (ITCZ) in MJJAS compared to NDJFM. The high probability ratios in northern hemispheric (NH) midlatitudes in local winter (NDJFM) are another discernible seasonal feature. Extreme precipitation in NH winter is primarily associated with large-scale systems, leading to a 5yRx1d distribution with relatively low variability. Due to strong projected winter warming in the high northern latitudes[40], the mean 5yRx1d trend in NH winter is large, and record-shattering probability is more sensitive to this mean trend due to the low variability. In NH summer, on the contrary, extreme precipitation is convective and variable, and there is no strong mean trend in 5yRx1d (see also Supplementary Fig. S11), damping the relative increase in record-shattering probability.

The spatial features of the empirical probability ratios are reproduced with very high similarity by the analytical probability ratios in Fig. 2c and d, corroborated by pattern correlations of over 0.9. Supplementary Fig. S10 shows that local temporal evolution of $P_{CCe}$ and $P_{CCa}$ is also very similar, confirming the agreement of the empirical and analytical absolute probability estimates. Figure 2e and f shows zonal means of empirical and analytical 5yRx1d probability ratios (solid lines) to enable easier quantitative comparison. Here we see, besides the remarkably high agreement of the empirical and analytical latitudinal patterns, that analytical probability ratios tend to be slightly lower. This might be explained by the limitations of GEV-fits in representing the highest end of the tail[41,42].

The agreement of empirical and analytical probability ratios indicates that non-stationary GEV distributions are helpful for estimating changing local record-shattering probabilities. This finding is important because empirical record-shattering probabilities can only provide robust results if very large model ensembles are available. The analytical GEV-based method provides the opportunity to estimate record-shattering behaviour in multi-model ensembles with fewer members per model (as many as needed to fit robust GEV distributions), which we demonstrate next.

## Global record-shattering precipitation probability in CMIP6 models

The finding that the analytical method approximates the empirical results closely is used to assess robustness of probability ratios across four other CMIP6 models with ensemble sizes between 40 and 10 members. Figure 2g and h (solid lines) demonstrate the similarity between zonal mean probability ratio patterns across models. The high probability ratios in the tropics flanked by low subtropical ones, and seasonal characteristics such as the location of the ITCZ and the high NH midlatitude probability ratios in local winter are consistent across models. Pearson correlation coefficients (PCC) of the zonal mean probability ratios of the models shown relative to CESM2-LE are significantly positive, but moderate with values ranging from 0.44 to 0.76 for MJJAS and 0.33 to 0.51 for NDJFM. Supplementary Section S4 features single- and multi-model probability ratio maps for the CMIP6 models.

An extensive multi-model assessment is outside the scope of this study, yet, the present analysis provides confidence that the precipitation record-shattering behaviour in CESM2-LE lies within the range of other CMIP6 models and that our analytical approach can be used for further characterisation of record shattering in precipitation. A comprehensive study by Abdelmoaty et al. (2021)[43] on CMIP6 model bias in extreme precipitation showed that CESM2 and also UKESM1-0-LL are among the top 10 of the 34 models analysed in terms of similarity to observations. CESM2 and UKESM1-0-LL have significantly smaller bias than ACCESS-ESM1-5 and the MPI models in mean, variability, skewness and kurtosis of extreme precipitation distributions. Particularly in the tropics, where mean and variability of precipitation is generally underestimated by climate models, CESM2 and UKESM1-0-LL feature a smaller underestimation. It is important to keep in mind, however, that the climate sensitivity of CESM2 is on the high end of the CMIP6 spectrum[44], and that SSP3-7.0 is a high warming scenario.

## Increasing variability key for increasing global-scale record-shattering probability

The analytical method for computation of $P_{CCa}$ enables us to disentangle the contributions of different changes in the 5yRx1d distribution to the changes in record-shattering probability. We do so by intervening on the GEV parameters we use to compute $P_{CCa}$. Note that we intervene after the fit has been made and do not fit a new GEV distribution. This is consistent in a GEV framework: subtracting the fitted $\mu$-timeseries from the data and refitting GEV distributions would result in the same $\sigma$-values. We break down record-shattering probability changes into a component due to location changes only by keeping $\sigma$ constant while allowing $\mu$ to vary, and a component due to scale changes only by keeping $\mu$ constant while allowing $\sigma$ to vary. We keep the parameters constant by setting their value to the 1850–1949 mean of the non-stationary parameter fit at all time steps (see 'Analytical record-shattering probabilities' for additional details). These components are shown in the dotted and dashed lines in Fig. 2e-h. Supplementary Fig. S13 shows global maps of the decomposed probability ratio contributions for CESM2-LE. Note that the two components are not expected to linearly add up to the total effect, since the record-shattering probability is not a linear function of $\mu$ and $\sigma$.

Varying $\mu$ and $\sigma$ individually yields latitudinal patterns of probability ratios that correlate strongly with the total probability ratios (Fig. 2e and f). A noteworthy difference between the components, however, is the larger contribution of the $\sigma$ component (dashed) compared to the $\mu$ component (dotted) across latitudes and seasons, apart from NDJFM in northern hemispheric midlatitudes. This suggests variability changes are the main

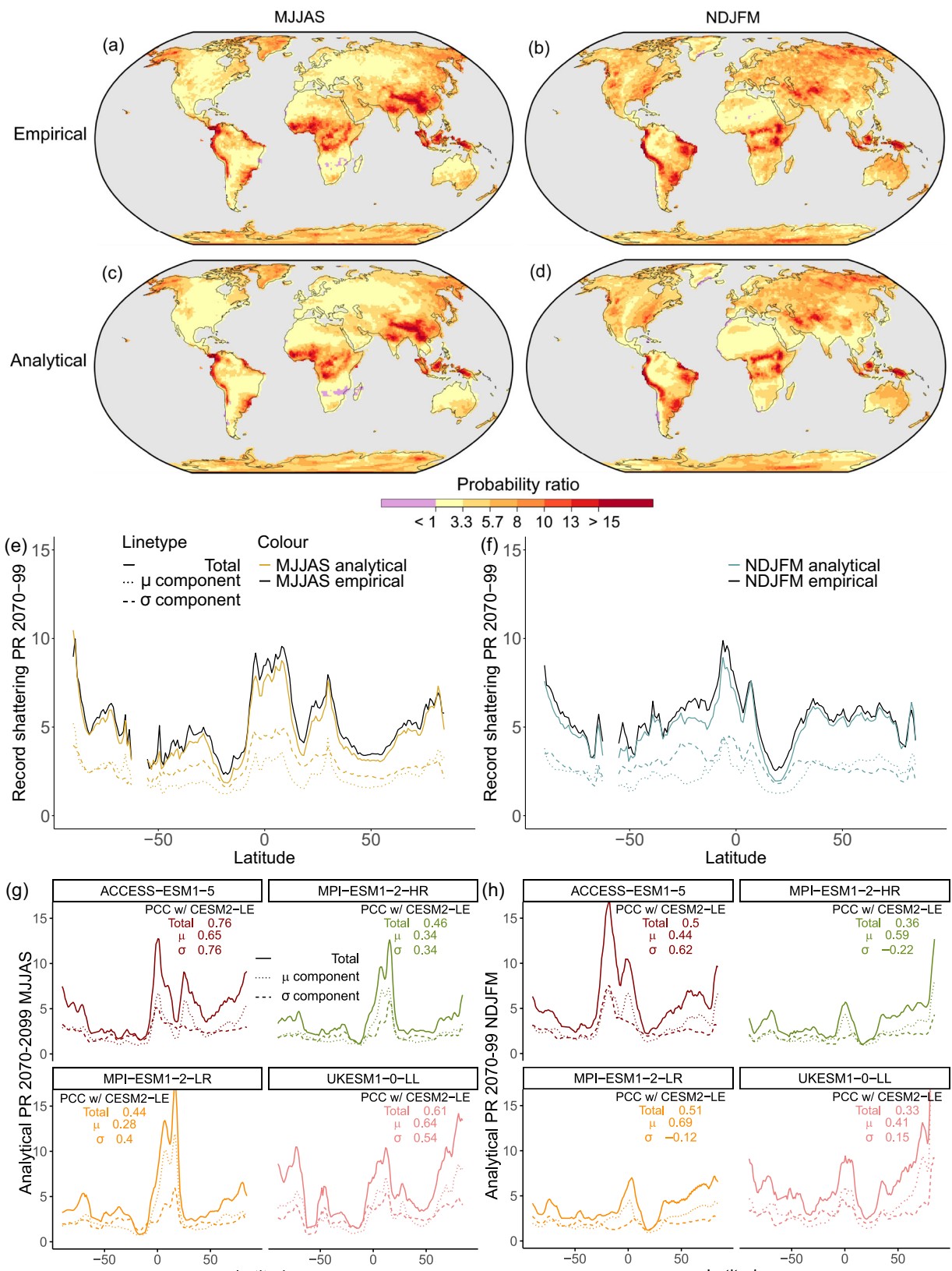

**Fig. 2 | Global 5-yearly Rx1d record-shattering probability ratios averaged for 2070–2099 in CESM2-LE and CMIP6 models forced with SSP3-7.0, for a record-shattering threshold of one pre-industrial standard deviation (1850–1949 average of the ensemble standard deviation) for extended hemispheric seasons.** CESM2-LE: (**a**) empirical MJJAS, (**b**) empirical NDJFM, (**c**) analytical MJJAS, (**d**) analytical NDJFM. **e**, **f** Zonal mean empirical and analytical probability ratios broken down into components due to GEV location parameter $\mu$ (mean) and scale parameter $\sigma$ (variability) for MJJAS and NDJFM. **g**, **h** CMIP6: Zonal mean analytical probability ratios for four CMIP6 models forced with SSP3-7.0 in MJJAS and NDJFM, including Pearson correlation coefficients (PCC) with CESM2-LE for the total signal, the $\mu$-trend component and the $\sigma$-trend component.

driver of the increasing chances of shattering precipitation records in a changing climate in CESM2-LE in most regions and seasons, a finding corroborated by the maps in Supplementary Fig. S13.

The breakdown for CMIP6 models in Fig. 2g and h shows that the $\mu$ and $\sigma$ contributions vary across models (see Supplementary Fig. S16 for global maps of component contributions). Model differences are particularly large in the tropics where probability ratios are the largest, underlining the need for improvement in modelling of extreme event severity and probability in vulnerable tropical regions. In extratropical regions, results are consistent and show that precipitation variability plays a considerable role for record-shattering probability increases. This finding is consistent with previous findings concerning the importance of variability for heavy (but not record-shattering) precipitation[45,46]. Overall, $\mu$ and $\sigma$ contributions of large, comparable magnitudes are present in all CMIP6 models, however, we note that $\sigma$ contributions are extraordinarily strong in CESM2-LE compared to the other models shown.

For the sake of completeness, we comment on the role of the shape parameter $\xi$. Since $\xi$ governs the tail properties of the distribution, its value affects record-shattering probabilities. For a constant $\xi$ with time, its primary effect is that increases in record-shattering probability for a given $\mu$- and $\sigma$-trend are damped for locations where $\xi$ is positive and large (i.e. approaching 0.5). This behaviour is seen since a heavy tail reduces the value of the last term in Eq. (2)—the probability that high 5yRx1d values have never been seen is smaller for heavier tails. The effect of $\xi$, however, is small. In this study, we do not further explore the effect of $\xi$ on record shattering.

The results shown here are sensitive to data and method choices. For example, the block length for which maxima are computed impacts the findings. The longer the blocks, the more dominant the effect of the $\mu$-trend, at the expense of the effect of the $\sigma$-trend. This is statistically intelligible, since longer block maxima, e.g. 10yRx1d, are selective samples from the tail of shorter block maxima and thus have lower variability. Additionally, the scaling of precipitation intensification with event rareness implies that the $\mu$-trend in 10yRx1d is likely to be stronger than in 5yRx1d[25–27]. Stronger $\mu$-trends and lower variability align with a larger contribution of the trend in $\mu$ to probability ratios for longer block maxima. Nonetheless, total probability ratios for 5yRx1d and 10yRx1d are very similar, indicating that these ratios are robust for record shattering in truly extreme events. Also, ensemble size can affect the results, however, as we show in Supplementary Section S5, sensitivity of the GEV-based method to ensemble size is small for ensembles of >20 members in CESM2-LE. Finally, Supplementary Section S6 shows similarities and differences in GEV-fits across CMIP6 models, demonstrating that these consistently translate to the corresponding model's record-shattering behaviour.

The probability ratios shown above are instructive to communicate the effects of climate change on record-shattering probability, but they do not show the actual probability of shattering a record. Absolute record-shattering probabilities are highly relevant for practical applications, hence the next two sections focus on regional absolute record-shattering probabilities. For a global overview, we show maps of absolute probabilities $P_{CC}$ and $P_{ref}$ in Supplementary Fig. S12. Note that these values hold within the scope of the analysis only, and are subject to model uncertainty and error.

**Regional changes in precipitation record-shattering probability**

In the following sections we quantify regional record-shattering probabilities in CESM2-LE in three selected regions (see Fig. 3). We assess the region in Belgium, The Netherlands, Luxembourg and Germany (BNLG) that was affected by heavy rainfall and floods in July 2021[33,34], and the region in Pakistan that was affected by floods resulting from heavy rainfall and glacier melt in June-August 2022[47–49]. We add the urban coastal zone near Lagos in Nigeria to the analysis, since this region lies in the fastest growing zone in the world—economically and in terms of population[50], and also features high probability ratios for record-shattering precipitation. An increasingly large number of people living in this urban zone might thus be exposed to ever more disastrous precipitation events.

The Lagos and BNLG regions consist of 9 grid cells, equivalent to roughly $400 \times 300$ km and $300 \times 300$ km, and the Pakistan region spans 8 grid cells and about $250 \times 400$ km (lon $\times$ lat). For regionally relevant insights, the quantity of interest is the probability of shattering a 5yRx1d record anywhere in the region. To this end, we pool the CESM2-LE record shattering counts of all grid cells in the region of interest, and determine for each time step the fraction of members that exhibit record shattering in any grid cell in the region (see 'Regionally pooled record-shattering probabilities'). This means that it does not matter in which or how many grid cells record shattering occurs, but only if it happens anywhere.

Pooling regional data for record-shattering probability estimates has several advantages relative to averaging or single grid cell approaches, that hold if the grid cells in the region of interest have strongly correlated 5yRx1d timeseries and a similar 5yRx1d distribution, i.e. if the region is climatically coherent. Firstly, determining the record shattering fraction over a larger sample containing data from multiple grid cells reduces noise. Secondly, averaging empirical record-shattering probabilities over multiple grid cells would compound the spatial extent and likelihood of record-shattering events, which complicates the interpretation of the results. Conversely, by pooling data for a physically meaningful region, we remove the sensitivity of the results to non-physical data attributes such as model grid cell size. Lastly, we argue that record-shattering probability for a climatically coherent region of a few 100 km across in both dimensions is a more policy-relevant metric than single grid cell probability since grid cell boundaries are arbitrary; event impacts act on spatial scales that are administratively and climatically meaningful. Naturally, there is a limit to the region size for which data pooling is useful: if the region is so large that record-shattering events at different ends are uncorrelated in terms of probabilities or impacts, the added value of pooling disappears. The regions we use are coherent, verified by spatial correlations averaged over all grid cell combinations within one region of 0.5–0.9.

Figure 3b–d shows the regional 5yRx1d timeseries. We focus on MJJAS since the flooding events in 2021 (BNLG) and 2022 (Pakistan) occurred in those months, and also probability ratios for Lagos are highest in those months due to the position of the ITCZ. Individual grid cell timeseries of the single member featuring the largest regional record-shattering event in the future are shown in grey, with red dots for record-shattering events. Record-shattering events far exceed the GEV-based interquartile range (blue shading) and often also the estimates of the 99th and 99.9th percentiles ($\approx$500 and 5000-year return levels). We also see that these events often affect multiple grid cells in the region (multiple red dots for one timestep) which further underscores the relevance of regional pooling. Increasing $\mu$ (blue line) and increasing variability (GEV interquartile range, blue shading) are seen for all regions, yet more pronounced for Pakistan and Lagos. Also, the number of record-shattering events in the 21st century is higher for Pakistan and Lagos than for BNLG, but the significance of this notion cannot be confirmed based on the shown data of one single member per region.

The empirical record-shattering probability (solid, black) in Fig. 3e–g shows a clear positive trend in all three regions. For BNLG (Fig. 3e), the record-shattering probability in a 5-year period in the local summer season increases from roughly 10% now to over 20% in 2100 under the SSP3-7.0 forcing scenario. In Pakistan, we see a threefold increase from about 10% now to over 30% in 2100. Even larger ratios are observed for Lagos, where 5yRx1d record-shattering probability increases to over 50% at the end of the century. This implies that it is more likely than not that anywhere in the $400 \times 300$ km region around Lagos, in any given 5-year period at the end of the century, the historical 5yRx1d record is broken with a margin comparable to the pre-industrial variability.

Note that the scenario used here, SSP3-7.0, is a high warming scenario, featuring about 4 °C warming above preindustrial temperatures in 2100—considerably more than projected with current policies. Note also that the record-shattering threshold (one pre-industrial standard deviation) is kept constant while climate change increases variability of 5yRx1d in the regions shown, meaning that the threshold no longer corresponds to one standard

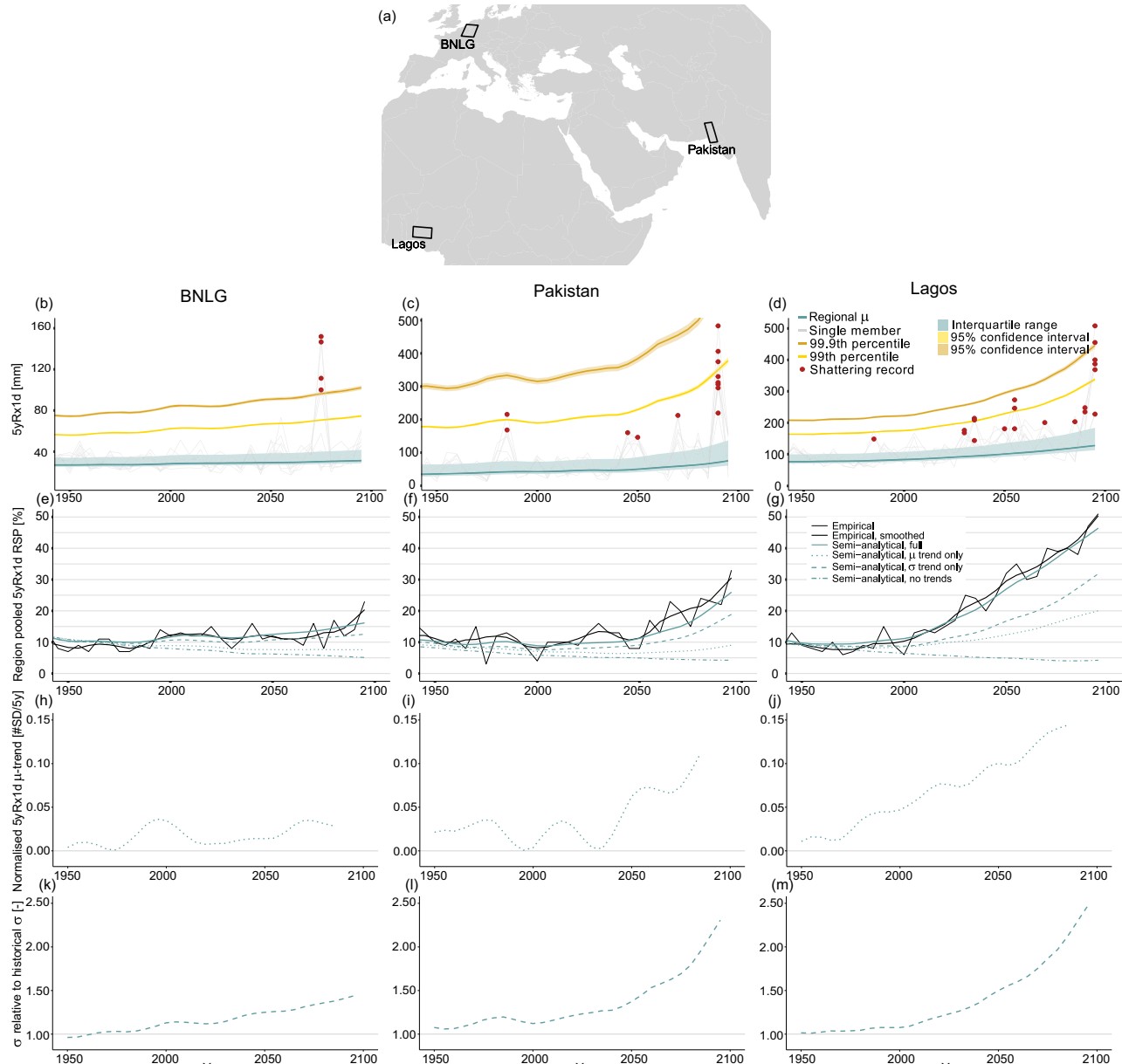

**Fig. 3 | Extreme precipitation characteristics and empirical and semi-analytical record-shattering probabilities for three climatologically coherent regions.**
**a** Regions analysed. **b–d** 5yRx1d timeseries of the ensemble member featuring the largest future record-shattering event (grey), each line is a grid cell in the region. Red dots show record shattering for each grid cell. The blue line and shading represent the location parameter $\mu$ of the region's non-stationary GEV-fit (one fit to data from all grid cells), and its interquartile range as a visually interpretable measure of variability, comparable to $\sigma$. Yellow/orange lines show the GEV-fit's 99th and 99.9th percentile, roughly equivalent to 500 and 5000-year return levels for 5yRx1d.

**e–g** Region pooled record-shattering probabilities (RSP) for 5yRx1d shattering anywhere in the region in a 5-year period, determined empirically from CESM2-LE (black), and semi-analytically from generated correlated GEV distributions (blue). Semi-analytical record-shattering probabilities due to $\mu$ and $\sigma$ trends only, as well as in stationary climate conditions ('no trends') are shown. **h–j** Normalised trend in $\mu$, defined as the linear trend of a 25-year window centred around the year in question, normalised by the historical 1850–1949 mean ensemble standard deviation ($\frac{1}{SD}\frac{\partial\mu}{\partial t}$). **k–m** Relative magnitude of $\sigma$ compared to its historical 1850–1949 mean.

deviation in the future climate. For Lagos MJJAS, the record-shattering threshold equals 25–30% of the historical 5yRx1d mean.

## Role of mean and variability for regional changes in precipitation record-shattering probability

In order to understand how local changes in the precipitation distribution govern the regionally different evolution of record-shattering probability, we require an analytical comparison based on distributional statistics as in Fig. 2. The analytical relationship in Eq. (2), however, cannot be used for regionally pooled probabilities since it applies to univariate timeseries.

Adapting the method to the multi-variate (multi-grid cell), spatially correlated timeseries used here is particularly cumbersome.

To nonetheless provide a semi-analytical comparison for regional record-shattering probabilities, and to enable separation of the contributions of trends in $\mu$ and $\sigma$, we generate synthetic spatially correlated GEV-distributed 5yRx1d samples based on GEV-fits to the regional CESM2-LE data (see 'Regionally pooled record-shattering probabilities'). The blue lines in Fig. 3e–g are based on the generated samples and show the component breakdown. The semi-analytical record-shattering probability estimates in Fig. 3e–g (solid blue) agree very well with the empirical estimates (solid

black). Also, the component breakdown provides valuable insights, as it corroborates the effect of increasing variability ($\sigma$, dashed) on increasing record-shattering precipitation probabilities on a regional scale. When only the $\sigma$-trend is considered, record-shattering probabilities increase from the year 2000 onward in all regions. The effect of the $\mu$-trend only ($\mu$, dotted) appears later and is weaker for all regions. The different features of the regional record-shattering probabilities can be explained by the characteristics of the changes in regional 5yRx1d. The metrics of importance for record shattering are the $\mu$-trend relative to the variability (normalised $\mu$-trend, $\frac{1}{SD}\frac{\partial\mu}{\partial t}$) and the magnitude of $\sigma$ relative to its historical value. These metrics are shown in Fig. 3h–m.

Extreme precipitation in the NH midlatitudes (BNLG) features small mean trends with warming relative to the tropics[29]. Yet, 5yRx1d variability is relatively high since we assess the local summer season which features primarily convective precipitation. Taken together, this leads to a small magnitude of the 5yRx1d normalised $\mu$-trend in BNLG (Fig. 3h). As a consequence, the $\mu$-trend alone is not enough to lead to a net increase in record-shattering probability, yet, it does increase the probability relative to the decrease with time seen in the stationary 'no trends' case. The skewed response of the precipitation distribution to warming leads to a relatively stronger increase in the most extreme events[25–27], in turn leading to a relatively larger and more consistent increase in $\sigma$ (Fig. 3k), which thus drives the increasing record-shattering probability in BNLG. Interestingly, the winter season in BLNG, featuring organised large scale precipitation and low variability, sees a predominant effect of the $\mu$ trend component, which can be explained by the higher magnitude of the normalised $\mu$-trend relative to climatological variability, see Supplementary Fig. S21. Wood (2023)[46] recently found similar results for the role of mean and variability trends for extreme precipitation in Europe.

Similar mechanisms hold for Pakistan, where 5yRx1d variability is high and increases strongly from 2000 onward (Fig. 3l), reflected in the record-shattering probability increase due to the $\sigma$-trend only. The normalised $\mu$-trend increases throughout the 21st century due to the relatively strong exponential intensification with temperature of extreme precipitation at subtropical latitudes[29]. As the normalised $\mu$-trend continues to increase (i.e. the $\mu$-trend slope becomes steeper) towards the end of the century (Fig. 3i), an increasing trend in record-shattering probability due to the $\mu$-trend only appears as well.

Lagos, finally, exhibits a clearly increasing normalised $\mu$-trend and increasing $\sigma$ throughout the 21st century (Fig. 3j, m). This increasing normalised $\mu$-trend—associated with high climatological precipitation and the strong scaling of extreme precipitation with temperature in the tropics[29]—means the full 5yRx1d distribution shifts upward fast enough to lead to increasingly frequent record shattering, even in the absence of increasing variability. Also for Lagos, however, the strong increase in $\sigma$ alone—associated with deep tropical convection and increasing atmospheric water vapour[37]—impacts record shattering more strongly than the changes in $\mu$ alone.

We can generalise the findings from Fig. 3 as follows: increasing variability directly translates to increasing record-shattering probabilities, whereas increases in the mean only result in increasing record-shattering probability in case of a large enough and steepening $\mu$-trend relative to natural variability (increasing normalised $\mu$-trend). A large enough but constant normalised $\mu$-trend (linear change in $\mu$), does not lead to increasing record-shattering probability but does slow down the decrease with time that we would see in a stationary climate. The total effect on record-shattering probability is a combination of the responses to both distributional changes.

The dominant role of $\sigma$ relative to $\mu$ apparent in Fig. 3 is partly because CESM2-LE shows strong 5yRx1d variability compared to other models (see Fig. 2g and h, and Supplementary Section S6) and may thus not be robust across model projections. Nonetheless, we can conclude that the distributional characteristics of extreme precipitation, in particular its high variability and skewed response to warming, result in a high sensitivity of the probability and intensity of unseen events to changes in variability.

As a final note, we point out that it is anticipated (albeit perhaps confusing) that the record-shattering probabilities at the end of the century in the stationary case (blue dash-dot in Fig. 3e–g) are still substantial ($\approx$5% in all regions). Since we assess 5yRx1d, the timeseries 1850–2100 contains 50 values, which equates to a stationary record breaking probability of $1/50 \approx 2\%$. The nonzero record-shattering margin decreases the probability, whereas the regional pooling increases it, leading to $\approx$5%.

## Analytical rationale

From the findings above a picture emerges in which record-shattering probability responds instantaneously to a positive trend in $\sigma$, whereas the sensitivity to the trend in $\mu$ is lower and appears with some delay. We formalise this theory using an analytical example that summarises the effects of different $\mu$- and $\sigma$-trend combinations.

Figure 4 shows how different $\mu$- and $\sigma$-trends affect record-shattering probabilities for an idealised GEV-distributed 5yRx1d-like dummy variable. We compare three $\mu$-trends, shown in Fig. 4a: no trend, a linear trend, and an exponential trend, corresponding to normalised $\mu$ trends of 0, constant, and increasing magnitude. Each $\mu$-trend is combined with three different $\sigma$ timeseries, constructed by taking 25% of the $\mu$-timeseries (see Eqs. (5) and (6)), shown in Fig. 4b. The assumed $\sigma/\mu$ ratio of 0.25 reflects the $\sigma/\mu$ ratio of the GEV-distributions fitted to the CESM2-LE 5yRx1d data (seasonal 5yRx1d global mean $\sigma/\mu = 0.27$). For reference, the corresponding normalised $\mu$-trend and $\sigma$-evolution for the Lagos region are shown. Shape parameter $\xi$ is kept constant at 0.1, a moderately positive value, which lies in the range of the fitted $\xi$ values. The $\mu$-$\sigma$ combinations result in the idealised 5yRx1d-behaviour shown in Fig. 4c–e (compare to Fig. 3a–c, where the blue line and shading show the same quantities for the actual 5yRx1d for the regions considered). See 'Analytical idealised example' for more details on the variable construction.

Figure 4f–h show the record-shattering probabilities corresponding to the combinations of different $\mu$- and $\sigma$-trends. Variability governs the short-term response of record-shattering probability: the larger the trend in $\sigma$, the steeper the initial increase in record-shattering probability. Both linear and exponential increases in $\sigma$ lead to increasing record-shattering probabilities, i.e. the time evolution of $\sigma$ directly translates to the time evolution of the record-shattering probability. The effects of the $\mu$-trend emerge more gradually. Compared to the stationary case (solid, light grey), the presence of a $\mu$-trend only (dotted, Fig. 4g and h) slows the decrease in record-shattering probability with time. In case of a linear $\mu$-trend (constant slope), we see that the record-shattering probability converges to a constant, nonzero probability as well (Fig. 4g). The magnitude of the constant record-shattering probability scales with the $\mu$-trend slope. In case of an exponential $\mu$-trend (increasing slope), we see that the record-shattering probability increases with time (Fig. 4h). This signifies that the slope of the trend in $\mu$ translates to the time evolution of the record-shattering probability. Note the different roles of $\mu$ and $\sigma$: linearly increasing $\mu$ with constant slope leads to constant record-shattering probability, whereas linearly increasing $\sigma$ with constant slope leads to increasing record record-shattering probability. Increasing $\mu$ leads to increasing record-shattering probabilities only in case of a steepening slope.

The different effects of $\mu$ and $\sigma$ on record shattering are a general property of the problem at hand and were documented previously for record-shattering temperatures[16]. The manifestation of these effects, however, depends on the distributional characteristics of the variable of interest. The relative effects of trends in $\sigma$ and $\mu$ depend on the natural variability of the variable assessed. Since the variability of extreme precipitation is high, normalised $\mu$-trends generally have moderate magnitudes and their effects take decades to centuries to manifest. For variables with smaller natural variability, normalised $\mu$-trends are larger and elicit responses in record-shattering probability earlier. The record-shattering probability evolution of extreme precipitation is particularly interesting due to the strong influence of variability, which is less present for temperature.

In summary, when $\mu$-trends and $\sigma$-trends are combined, we see a response made up of two parts, where the $\sigma$-trend dominates an initial surge

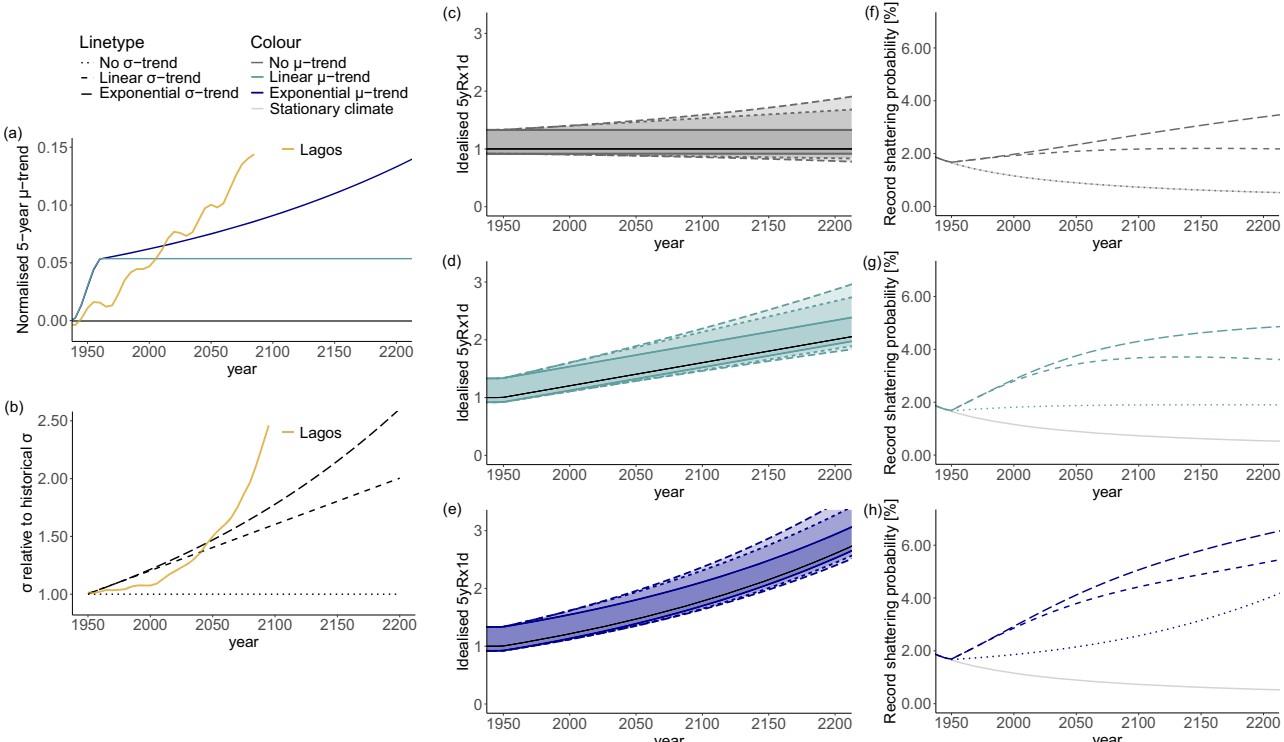

**Fig. 4 | Dependence of analytical record-shattering probability for idealised GEV-distributed 5yRx1d-proxy on $\sigma$ and $\mu$. a** Evolution of the different assessed normalised $\mu$-trends, defined as the trend in $\mu$ relative to the initial (historical) standard deviation $(\frac{1}{SD}\frac{\partial\mu}{\partial t})$; (**b**) Evolution of the different assessed $\sigma$-trends relative to initial (historical) $\sigma$ ($\frac{\sigma_t}{\sigma_1}$ with $t$ = time); **c–e** $\mu$ and GEV interquartile range of the idealised 5yRx1d-proxy for all combinations—compare to Fig. 3a–c; (**f–h**) record-shattering probabilities for the different combinations of $\mu$ and $\sigma$ trends. A longer time period up to year 2200 is shown for the sake of clarity and convergence of the response.

in record-shattering probability, but fades over time to converge to the slope prescribed by the $\mu$-trend slope, albeit at a level influenced by the slope of the $\sigma$-trend. Thus, local precipitation record-shattering probability changes can be expected to follow the local $\sigma$-evolution in the near-term (decades), followed by long-term behaviour (multi-decade—centuries) governed by the local variability-normalised $\mu$-trend slope.

## Conclusions

Record-shattering weather events—events that exceed the historical record by a large margin of e.g. one pre-industrial standard deviation—can be devastating to communities and ecosystems since they lie far outside the range of what society has experienced and adapted to in history. We show that climate models project an increasing probability of occurrence of record-shattering precipitation in a changing climate. The probability of shattering 5yRx1d records in CESM2-LE under scenario SSP3-7.0 increases for most of the global land relative to a world without climate change. In some tropical regions, the local record-shattering probability ratio at the end of the 21st century is as large as 15, meaning the record-shattering probability is 15 times larger than in a world without climate change.

Record-shattering probabilities determined empirically from the 100-member ensemble CESM2-LE data can be reproduced analytically, using theory and non-stationary GEV distributions fitted to the model data. The analytical method, being less sensitive to ensemble size, can be used to estimate record-shattering probabilities in smaller model ensembles. This reveals that four different CMIP6 models exhibit qualitatively similar behaviour to CESM2-LE, but differ in precise projections of precipitation distribution changes. A more extensive multi-model study is needed to quantify the exact magnitude of changes and uncertainties in probabilities of record-shattering precipitation across models and under different emission scenarios.

We quantified regional record-shattering probabilities in CESM2-LE under the relatively high warming scenario SSP3-7.0, defined as the probability that records are shattered anywhere in regions of about 400 km in cross section. More than doubling of the 5yRx1d record-shattering probability by the end of the century relative to the present is projected for several regions in both the tropics and the midlatitudes. Some tropical regions, where the probability increase relative to a world without climate change is highest, become subject to a chance of roughly 50% of shattering the local 5yRx1d record in any given 5-year period by the end of the century in the climate model and scenario we study. This applies, for example, to the economically and demographically fastest-growing region of the world near Lagos, Nigeria.

Our most important finding, consistent across model simulations and analytical approximations, is that increasing variability of local extreme precipitation leads to a fast positive response of record-shattering probability, and steepening mean trends in local extreme precipitation lead to long-term positive trends in record-shattering probabilities. In a changing climate, both of these conditions are present in most land regions given the exponential relationship of extreme precipitation with temperature[20–23], and the skewed character of precipitation change where intensification scales with event rareness[25–27]. The global climatology of precipitation—with most intense extremes and high variability in the tropics and monsoon regions—implies that the changes will emerge first and strongest in these regions, which are often vulnerable to natural disasters.

The quantitative probabilities reported in this study are strongly dependent on choices made in the analysis, such as block length and region dimensions, and also on the warming scenario and climate models used. CESM2, the core climate model used in this study, has been shown to reproduce the distribution of extreme precipitation comparatively well[43]. The high climate sensitivity of CESM2, however, needs to be noted when interpreting the results[44]. Furthermore, global models do not resolve all

processes that drive extreme local precipitation. Higher resolution models would likely simulate more extreme local precipitation intensities over heterogeneous terrain, but also higher variability[51], thus making it hard to predict how record-shattering probability depends on model resolution. The purpose of this study, however, is to present a conceptual framework to improve understanding and estimation of the probabilities of high-impact precipitation events. Albeit quantitative estimates will vary depending on the context, we believe the framework we present can prove to be invaluable for future assessment of record-shattering precipitation events.

Extreme event occurrence probability is, in general, highly sensitive to changes in variability[52]. This is strikingly relevant for extreme precipitation events since precipitation is projected to see comparatively large changes in variability. To improve adaptation-relevant policy information, it is important to constrain projections of changing variability of impact-relevant indices such as extreme precipitation in a changing climate.

## Data and methods

### Data

Observational and reanalysis data used in Fig. 1 stems from E-OBS and ERA5, both at 0.25° resolution[36,53]. The model data used in the core part of the analysis stems from the Community Earth System Model 2 Large Ensemble Community Project 2 (CESM2-LE)[30], a 100-member initial condition ensemble at a nominal resolution of 1°, forced with historical forcing in 1850–2014, followed by SSP3-7.0 up to 2100[32]. For the analytical GEV-based estimates of record shattering in CMIP6 models ('Global record-shattering precipitation probability increases relative to 140 stationary climate' and Supplementary Section S6), the four available CMIP6 models with relatively high resolution (at least 1.875°) and at least 10 individual members covering the full 1850–2100 record with historical and SSP3-7.0 forcing, namely ACCESS-ESM1-5 (40 members), MPI-ESM1-2-HR (10 members), MPI-ESM1-2-LR (30 members), and UKESM1-0-LL (13 members), were used[54–61].

Annual maxima of daily precipitation (Rx1d) were extracted using Climate Data Operators (CDO)[62]. For the CMIP6 models, Rx1d values were extracted on their respective native grids, after which the fields were conservatively regridded onto the CESM2-LE-grid (CDO remapcon2[62,63]) for the sake of comparison. Note that this means CMIP6 data were regridded onto higher resolution grids for all models except MPI-ESM1-2-HR, which introduces biases. However, given that the CMIP6 analysis serves as a qualitative comparison, this is not critical. For all models, data were land-masked using a landmask on the CESM2-LE-grid. 5-year block maxima (5yRx1d) were computed in the analysis routines in R[64] by extracting the maximum Rx1d value in chronological, non-overlapping, 5-year windows. The value assigned to e.g. the year 2000 thus applies to the window 2000-2004.

### Meteorological event analysis

For the Rx1d anomaly timeseries shown in Fig. 1a, anomalies are determined relative to the mean of the observational coverage period from 1950–2022 for both E-OBS and CESM2-LE data, to ensure both timeseries are centred around the same baseline. The construction of the GEV-statistics is explained in the next section.

Figure 1b–i show daily values of meteorological variables of interest for extreme precipitation in the region in the marked red box. For E-OBS and ERA5, the data corresponds to July 14th, 2021. For the simulated future event, the event date is July 21st, 2077. For precipitation and vertically integrated water vapour transport (IVT) (b and f, and d and h), the daily absolute value is shown. For vertical updraft, sea level pressure and geopotential height (c and g, and e and i) daily anomalies are shown. The anomalies are determined with respect to the multi-year daily mean of the full record, i.e. 1950–2022 for the observational data and 1850–2100 for the CESM2-LE data. This means the reference period for the two events is different, and also, the events take place in a different climate background. Therefore, absolute magnitude of the daily values and anomalies are not

comparable, and instead the focus lies on the patterns and the meteorological state they represent.

IVT is provided in ERA5, but computed for CESM2-LE using Eq. (3), where $\overline{IVT}$ is a vector containing the eastward and northward IVT components in $\text{kg m}^{-1}\,\text{s}^{-1}$, $p$ refers to pressure in $\text{N m}^{-2}$ (or equivalently, Pa), $q$ to specific humidity in $\text{kg kg}^{-1}$, $\overline{U}$ to the velocity vector with eastward component $u$ and northward component $v$ in $\text{m s}^{-1}$ (both functions of $p$) and $g$ is the gravitational constant in $\text{N kg}^{-1}$ (or equivalently, $\text{m s}^{-2}$). The integral in (3) is evaluated as a discrete sum across the pressure levels spanning the full atmospheric column from 1000 hPa to 2 hPa.

$$\overline{IVT} = \frac{1}{g} \int_p \overline{U} q \, dp \qquad (3)$$

### Extreme value analysis

Generalised extreme value (GEV) distributions are used throughout the analyses in this study. We use the R-package extRemes[65] to fit GEV distributions. GEV distributions describe distributions of block maxima, such as annual or multi-annual maxima, and are defined by three parameters: the location parameter $\mu$, scale parameter $\sigma$, and shape parameter $\xi$ represent centre, spread, and tail characteristics. The fitted GEV distributions provide continuous descriptions of the discrete (simulated) sample data. These allow estimation of probabilities of very unlikely events (long return periods) which cannot reliably be determined from the sample data directly due to finite sample size.

In our study, we employ GEV distributions in various ways for the different results shown. Firstly, to arrive at the observations-model comparison shown in Fig. 1a, we fitted stationary GEV distributions to the 71 years (1950–2020) of observational E-OBS[36] Rx1d data preceding the 2021 European flooding event, as well as to the 71 years of single-member CESM2-LE data preceding the 2077 precipitation event. We averaged Rx1d data over the region $4–8°\text{E} \times 49–52°\text{N}$ (the red boxes in Fig. 1b–i) for both observational and simulated data before fitting the GEV distributions, to smooth out differences due to the different spatial resolutions of the observations and the simulations. The GEV-fits are determined using L-moments estimation and parametric bootstrapping for the confidence intervals within the *fevd()*-function of the extRemes R-package[65]. This results in the $\mu$ values and return levels shown in Fig. 1a.

Secondly, we fitted grid cell specific non-stationary GEV distributions to the CESM2-LE and other CMIP6 model 5yRx1d data. Non-stationarity implies that the GEV distribution is allowed to vary in a user-defined way. In our setup, $\mu$ and $\sigma$ vary as a linear function of covariates $g_\mu(t)$ and $g_\sigma(t)$, which are functions of time. This setup allows us to account for effects of climate change on the GEV distribution. $\xi$ is kept constant with time since estimation of $\xi$ is associated with high uncertainty[66], and it is valid to assume that the tail characteristics of local extreme precipitation distributions remains reasonably constant with (moderate) climate change. Supplementary Section S2 validates our GEV parameterisation. This leads to the following definition of the GEV parameters:

$$\begin{cases} \mu(t) = \mu_0 + \mu_1 g_\mu(t) \\ \sigma(t) = \sigma_0 + \sigma_1 g_\sigma(t) \\ \xi = \xi_0 \end{cases} \qquad (4)$$

We use covariates that are closely related to the corresponding GEV parameter, namely, $g_\mu(t)$ is the ensemble mean and $g_\sigma(t)$ is the ensemble standard deviation of 5yRx1d at the grid cell of interest. Both covariates are smoothed with a lowess filter with a span of $\approx 12\%$, which corresponds to a $\approx 31$-year rolling mean given our record of 251 years (in practice, the equivalent of 31 year smoothing for 5yRx1d employs a rolling window of 6 datapoints; 5 years * 6 = 30 years)). The five parameters $\mu_0, \mu_1, \sigma_0, \sigma_1, \xi_0$ are determined using maximum likelihood estimation within the *fevd()*-function of the

extRemes R-package, and a normal approximation for confidence intervals, which is justified for multi-member ensemble sample sizes[65].

The covariate-based fitting approach constrains the freedom of the parameters' evolution in time since they can only vary linearly with the covariate. For validation of this GEV-approach, we fitted separate GEV distributions to the $N = 100$ sample (CESM2-LE members) at every time-step individually, allowing total freedom of the time evolution of the GEV parameters. The resulting GEV parameters were in very close agreement with those found using the covariate-based non-stationary GEV-fit, validating that the latter represents the time evolution of the 5yRx1d distribution well. Relative to timestep-specific GEV-fits, the covariate-based approach has as a major advantage that the full sample—all timesteps—are used for fitting the distribution, increasing its accuracy. In addition, the number of estimated parameters is much lower (5 instead of $251 \times 3$), making the statistical model simpler and more tractable.

Supplementary Section S2 addresses the goodness of fit of the GEV distributions that result from the above procedure. We validate that the setup with non-stationary $\mu$ and $\sigma$ but stationary $\xi$ is the optimal setup, based on a trade-off between accuracy and tractability.

We also fitted grid cell specific stationary GEV distributions to the CESM2-LE and CMIP6 5yRx1d data from 1850–1949, which describe the reference 5yRx1d distribution in an unchanged climate. For this we use L-moments estimation within the *fevd()*-function of the extRemes package[65] for higher robustness[67].

Lastly, we fitted non-stationary GEV distributions to regionally pooled CESM2-LE data for Fig. 3b–d. The regional GEV distributions are determined in the same way as the grid cell specific GEV distributions described above, however, the data from all grid cells in the region of interest is pooled together per timestep and member. The covariates are also determined based on the regionally pooled data, meaning $f_\mu(t)$ and $f_\sigma(t)$ are now the ensemble mean of the spatial mean and the ensemble standard deviation of the spatial mean. For Fig. 3e–g, GEV distributions are fitted to individual grid cells, see also below. The goodness of fit for these GEV distributions is also validated in Supplementary Section S2.

### Empirical record-shattering probabilities

Record shattering is path dependent: whether the latest observation is a record depends on the historical evolution of the observed variable in question. To compute record-shattering probabilities directly (empirically) in CESM2-LE, we thus have to treat each ensemble member individually. We determine global trends in 5yRx1d record shattering between 1950–2100 under historical + SSP3-7.0 forcing[32]. We evaluate at every timestep whether the current value exceeds the maximum of all preceding values plus the user-defined record-shattering threshold $c_{rs}$, for each member and each grid cell. We choose the grid cell specific record-shattering threshold to be constant with time, and equal to the 1850–1949 mean of the ensemble standard deviation, reflecting pre-industrial natural variability. Climate change affects the variability and thus the standard deviation of extreme precipitation[68], meaning that our constant record threshold does not reflect natural variability throughout the period analysed. Using a varying record-shattering threshold, however, would complicate the interpretation of the results. Also in practical terms, the past natural variability is likely to have informed regional policies, and thus is a relevant record-shattering threshold to use for adaptation questions.

The magnitude of the chosen threshold for record shattering affects the results. For higher thresholds (more extreme records) the magnitude of probabilities decrease, but the qualitative behaviour of record-shattering probabilities over time is insensitive to the magnitude of the threshold. From a practical point of view, the precise shattering threshold is of limited importance: any considerable margin poses a greater risk for high damage since infrastructure is often not designed for such events.

The expression we evaluate for each member, grid cell, and timestep is given by Eq. (1). We use 5yRx1d as the example variable since most of our analysis focuses on 5yRx1d, but this expression would be the same for any block length. If this expression is true for a certain member–timestep–grid

cell combination, a record-shattering event is recorded. The empirical (ₑ) record-shattering probability with climate change (CC) $P_{CCe}(j, t)$ is then defined as the fraction of members featuring a record-shattering event at time $t$ and grid cell $j$.

The empirical record-shattering probability used to compute the empirical probability ratios shown in Fig. 2a–b corresponds to the 2070–2099 average of $P_{CCe}$. To obtain the empirical probability ratios, $P_{CCe}$ is divided by the 2070–2099 average of the analytical record-shattering probability in an unchanged reference climate $P_{ref}$, which is defined in the next section.

### Analytical record-shattering probabilities

In order to determine analytical (ₐ) record-shattering probabilities in a changing climate $P_{CCa}$ and in a stationary reference climate $P_{ref}$, non-stationary GEV distributions for the period 1850–2100, and stationary GEV distributions for the period 1850–1949 are fitted for each individual grid cell, as described above. The thus obtained GEV distributions' probability density function and cumulative distribution function are subsequently used in Eq. (2) to determine the analytical record-shattering probabilities. Dividing $P_{CCa}$ by $P_{ref}$ gives the analytical record-shattering probability ratios, and dividing $P_{CCe}$ (see previous section) by $P_{ref}$ gives the empirical record-shattering probability ratios.

The analytical approach allows for separation of the effects of trends in $\mu$ and $\sigma$ on the analytical record-shattering probability ratios. To achieve this, we modify the GEV distributions fed into Eq. (2) and obtain decomposed values for $P_{CCa}$ due to a trend in $\mu$ or $\sigma$ only. To determine $P_{CCa}$ due to the trend in $\mu$ only, we let $\mu$ vary according to the non-stationary fit and keep $\sigma$ constant at its 1850–1949 average. Vice versa, $\mu$ is kept constant while $\sigma$ is allowed to vary to determine $P_{CCa}$ due to the trend in $\sigma$ only. Dividing these trend-separated $P_{CCa}$s by $P_{ref}$ gives the analytical record-shattering probability ratios due to the trend in either $\mu$ or $\sigma$.

Note that the mean of a GEV-distributed variable depends on $\mu$ and $\sigma$. This implies that it is impossible to intervene on only one of these parameters without (unintentionally) affecting the mean. For the case where we keep $\sigma$ constant at its historical value and let $\mu$ vary, the mean of the post-intervention distribution is lower than the mean of the original distribution; for the case where we keep $\mu$ constant while $\sigma$ is allowed to vary, the mean of the post-intervention distribution increases with increasing $\sigma$ even though $\mu$ is constant. It is thus important to view the trend-separation procedure in the GEV-context, i.e. in terms of location and scale parameters.

We apply the same procedure to obtain analytical (decomposed) record-shattering probability ratios for the other four CMIP6 models discussed in 'Global record-shattering precipitation probability increases relative to 140 stationary climate' and Supplementary Section S6.

The robustness of the estimates of analytical record-shattering probability hinges on the robustness of the GEV-fits underlying these estimates. Supplementary Section S5 shows that the the GEV estimates are robust for ensemble sizes between 100 and 5 members, albeit uncertainty increases as ensemble size decreases, particularly for ensemble sizes below 20. Supplementary Section S6 shows the variation in GEV parameter estimates between CESM2-LE and the four other CMIP6 models we discuss.

### Regionally pooled record-shattering probabilities

Empirical regionally pooled record-shattering probability at time $t$ is, as explained in the main text, defined as the probability of a record-shattering event occurring anywhere in a confined region of interest at time $t$. In this approach, record shattering is initially evaluated per grid cell with grid cell specific record-shattering thresholds. Then we construct the regionally pooled record-shattering timeseries by determining, for each timestep, the fraction of members for which any grid cell in the region exhibits record shattering. We choose regions of a size small enough to ensure high correlation of the precipitation records in the different grid cells since the goal of regional pooling is to assess policy-relevant record-shattering probabilities in a climatically coherent region. The regionally pooled record-shattering probability is sensitive to the spatial correlation in the region. In a region

with $n$ perfectly correlated grid cells with a shattering probability of $p_i$ each, shattering would always happen in all grid cells at the same time. This means that the probability that shattering happens anywhere in the region is $p_i$ as well, and is thus equivalent to the probability averaged over the grid cells. In a region with $n$ perfectly uncorrelated grid cells with a shattering probability of $p_i$ each, the probability that record shattering happens anywhere is $1 - (1 - p_i)^n$. (In practice, this is very close to the sum of the individual grid cell probabilities $n p_i$, since $p_i$ is very small and thus grid cells would hardly ever shatter simultaneously if they are not correlated: if $p_i << 1$, $p_i^k \approx 0$ with $k > 1$, and $1 - (1 - p_i)^n \approx n\,p_i$.) In our case, the spatial correlation within regions is high but not perfect. This means that shattering normally happens in a few of the grid cells within a region, and the probability that any of the grid cells shows record shattering is higher than the average of the individual grid cell probabilities but lower than the sum of the individual grid cell probabilities. Assessing record shattering anywhere in a region removes influence of the spatial dimension of the event on the probability, making it more interpretable. However, it is important to limit region size, since the probability of record shattering anywhere in a large region with low spatial correlation becomes meaninglessly large.

The results of the regional pooling approach described above, leading to the record-shattering probabilities displayed in Fig. 3, cannot be compared to the analytical solution presented in above because the 5yRx1d timeseries in the grid cells of a region are spatially correlated.

Hence, we introduce a more tractable, semi-analytical method to provide verification of the empirical regionally pooled record-shattering probability estimation. We determine estimates of regionally pooled record-shattering probabilities by generating large samples of synthetic 5yRx1d values based on GEV distributions fitted to the grid cells in the region of interest and the grid cells' spatial correlation. The sample-generation method follows the steps below:

1. Fit a non-stationary GEV distribution to the 5yRx1d data of each grid cell in the region
2. Translate GEV-distributed grid cell data to normally distributed data
    (a) Determine the quantiles of all 5yRx1d entries based on their respective GEV-fits
    (b) Input quantiles in standard normal distribution for each grid cell
3. Determine spatial covariance matrix based on normally distributed data
4. Generate $10^4$ normally distributed synthetic timeseries per grid cell, with spatial correlation prescribed by covariance matrix determined above
5. Translate normally distributed synthetic data back to original GEV-distributed data
    (a) Determine the quantiles of generated values in standard normal distribution
    (b) Input quantiles in grid cell specific non-stationary GEV distributions from step 1

The result of these steps is a regional set of idealised GEV-distributed and spatially correlated 5yRx1d timeseries. We then compute the regionally pooled record-shattering probabilities in this synthetic sample in the empirical way, as described at the start of this section. In order to isolate the effects of $\mu$ and $\sigma$ trends—as shown in Fig. 3c–d—we intervene on step 5b listed above. Namely, instead of inputting the generated quantiles into the original, non-stationary GEV distributions, we input them into modified GEV distributions in which either $\mu$, $\sigma$, or both $\mu$ and $\sigma$ are set to their (constant) average 1850–1949 values. In this way, we separate changes in regional record-shattering probability due to the trend in $\mu$ only or $\sigma$ only and can determine the reference record-shattering probability in a stationary climate (no trends).

In statistical terms, the above step process describes the generation of multivariate data with marginal GEV distributions and a Gaussian dependence copula. This method is not fully consistent with extreme value theory (EVT), since EVT would prescribe the use of extreme-value dependence copulas. However, it has been shown that these can be too rigid in practical

applications, primarily due to strong constraints on tail dependence[69]. Besides, the generation of multivariate GEV distributions and the choice of a suitable extreme-value dependence copula is not straightforward and presents with many caveats[69,70]. Gaussian dependence copulas are more flexible and easier to handle and are considered the most suitable for our purpose, which is not to generate highly specific and accurate multivariate GEV distributions but to obtain a data sample that is representative enough of the 5yRx1d behaviour in the region of interest.

To validate the data generation method, we ensured that all covariance matrices in the standard normal space are positive definite. This is necessary to enable generation of correlated samples. Secondly, an important criterion for validity of the method is a satisfactory goodness of fit of the GEV distributions fitted to the 5yRx1d data. A validation of goodness of fit is provided in Supplementary Section S2. In addition, we confirmed congruence of the correlation matrices of the original CESM2 5yRx1d data and the generated 5yRx1d data to verify that the generation process via the standard normal yields a sample with similar spatial dependencies as the original sample, also in the GEV space. Supplementary Fig. S20 shows the correlation matrices of the original MJJAS 5yRx1d data, generated MJJAS 5yRx1d data, and their difference for the three regions we focus on.

## Analytical idealised example

The analytical example shown in Fig. 4 shows the effects of trends in GEV-parameters $\mu$ and $\sigma$ on record shattering in a GEV-distributed 5yRx1d-like variable. Assuming three possible trend-shapes for both $\mu$ and $\sigma$—no trend, a linear trend, and an exponential trend—leads to nine possible (qualitative) $\mu$-$\sigma$ trend combinations that cover a range of plausible regional extreme precipitation distribution changes we might observe with climate change.

We assume an idealised, dimensionless 5yRx1d variable, trends are prescribed in terms of relative change. The idealised initial distribution has a location parameter $\mu$ of 1 and a scale parameter $\sigma$ of 0.25, which is representative of the relative magnitude of $\mu$ and $\sigma$ of the GEV-distributions fitted to the CESM2-LE 5yRx1d data (seasonal 5yRx1d global mean $\sigma/\mu = 0.27$). $\mu$ and $\sigma$ are varied to assess the effects of different combinations of distributional changes. $\xi$ is kept constant at 0.1, a moderate value within the range of the GEV-distributions fitted to the CESM2-LE 5yRx1d data (seasonal 5yRx1d global mean $\xi = 0.07$). We chose the idealised trends such that they represent a time evolution plausible for regional extreme precipitation changes, yet, they are on the high side for the sake of signal strength.

Over most of the global land, extreme precipitation changes are primarily thermodynamically driven, via changes in the water holding capacity of the atmosphere with changing temperature[24]. Hence, we assume a warming trend (timeseries of temperature anomaly) and compute the corresponding extreme precipitation $\mu$ and $\sigma$ trends based on thermodynamic scaling. The observed warming over the global land in the past 100 years lies around 1.5K[71]. For illustrative purposes and since regional (and seasonal) temperatures can increase at higher rates than global mean land temperature, we assume a higher linear warming rate of about 4K per 100 years in our idealised example. For simplicity, we construct a temperature anomaly timeseries consisting of two linear segments; a segment where the temperature anomaly is constant at 0 from 1850-1950, and a linear increase with a constant rate of 4K per 100 years thereafter.

The theoretical Clausius-Clapeyron relationship prescribes an exponential increase in water holding capacity of the atmosphere with warming of ≈7%/K. For very extreme precipitation, such as 5yRx1d, increases at rates higher than the Clausius-Clapeyron rate of 7% have been postulated[38,72], hence, we assume a 10% increase in 5yRx1d per degree of regional warming in the idealised example.

Since record-shattering temperatures have been shown to be driven by warming rate rather than warming level[16], we assess the role of the rate of change for record-shattering precipitation as well. To this end, we compare the effects of an exponential 5yRx1d trend to a linear trend of 10%/K in $\mu$. To systematically analyse the effects of (combined) trends in $\mu$ and $\sigma$, we apply linear and exponential trends to the $\sigma$ parameter as well, defined as functions of the $\mu$-trends. Equations (5) and (6) show the combinations of $\mu$ and $\sigma$

trends used, also shown in Fig. 4, as a function of temperature change $\Delta T$, where the subscripts 'o', 'lin', and 'exp' refer to no, linear, and exponential trends.

$$\begin{cases} \mu_{o} = 1 \\ \mu_{lin} = 1 + 0.1 \cdot \Delta T \\ \mu_{exp} = 1.1^{\Delta T} \end{cases} \quad (5)$$

$$\begin{cases} \sigma_{o} = 0.25 \cdot \mu_{o} \\ \sigma_{lin} = 0.25 \cdot \mu_{lin} \\ \sigma_{exp} = 0.25 \cdot \mu_{exp} \end{cases} \quad (6)$$

All combinations of $\mu$ and $\sigma$ trends are used to describe time-evolving GEV distributions for which the record-shattering probabilities are computed using Eq. (2), resulting in Fig. 4. The record-shattering threshold used here is the 1850–1949 mean standard deviation, as in the main analysis. The standard deviation is computed analytically from the GEV parameters.

The normalised slope of the $\mu$-trend is predictive of the effect of $\mu$ on record-shattering probability. It is computed by dividing the 5-year trend in $\mu$ by the initial standard deviation (which is the same as the 1850–1949 pre-industrial standard deviation, and thus also the record-shattering threshold).

## Data availability
CESM2-LE data used in this study are publicly available at https://www.earthsystemgrid.org/dataset/ucar.cgd.cesm2le.output.html All original CMIP6 data used in this study are publicly available at https://esgf-node.llnl.gov/projects/cmip6/. ERA5 data are publicly available from the Copernicus Climate Data Store at https://cds.climate.copernicus.eu/#!/home We acknowledge the E-OBS dataset from the EU-FP6 project UERRA and the Copernicus Climate Change Service, and the data providers in the ECA&D project (https://www.ecad.eu).

## Code availability
Preprocessed data and the code to compute the values and produce the figures in this paper are available at https://doi.org/10.3929/ethz-b-000684058; additional code is available upon request.

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

## Acknowledgements

We would like to thank Marius Egli, Ned Glick, Vincent Humphrey, and Raphael Huser for helpful discussions. We thank Urs Beyerle, Lukas Brunner and Ruth Lorenz for the preparation and maintenance of CMIP6 data. We acknowledge the World Climate Research Programme's Working Group on Coupled Modelling, which is responsible for CMIP, and we thank the climate modelling groups for producing and making available the model output. For CMIP, the U.S. Department of Energy's Program for Climate Model Diagnosis and Intercomparison provides coordinating support and led development of software infrastructure in partnership with the Global Organization for Earth System Science Portals. We acknowledge the CESM2 Large Ensemble Community Project and supercomputing resources provided by the IBS Center for Climate Physics in South Korea. We acknowledge the E-OBS dataset from the Copernicus Climate Change Service (C3S, https://surfobs.climate.copernicus.eu) and the data providers in the ECA&D project (https://www.ecad.eu), and the European Centre for Medium Range Weather Forecasts (ECMWF) for providing ERA5 reanalysis data. The analysis was carried out in R (R Core Team, 2022), thus we thank all contributors for the numerous R packages crucial for this work, in particular the extRemes package. JZ acknowledges funding from the Swiss National Science Foundation within the project 'Understanding and quantifying the occurrence of very rare climate extremes in a changing climate' (Grant 200020_178778).

## Author contributions

Iris de Vries contributed to conceptualisation, method development, method implementation and data analysis, and writing and visualisation. Sebastian Sippel, Erich Fischer, and Joel Zeder contributed to conceptualisation, method development and text improvements, Reto Knutti contributed to conceptualisation, text improvements and funding acquisition.

## Competing interests

The authors declare no competing interests.
