## [Peer Review File · Communications Earth & Environment]

Increasing extreme precipitation variability plays a key role in future record-shattering event probability

Corresponding Author: Dr Iris de Vries

Version 0:

Decision Letter:

Dear Ms de Vries,

First of all, please allow me to sincerely apologise once again for the long delay in sending a decision on your manuscript titled "Surge in future record shattering precipitation driven by increasing variability". It has now been seen by 2 reviewers, and we include their comments at the end of this message. The reviewers find your work of interest, but some apparently important points are raised. We are interested in the possibility of publishing your study in Communications Earth & Environment, but would like to consider your responses to these concerns and assess a revised manuscript before we make a final decision on publication.

We therefore invite you to revise and resubmit your manuscript, along with a point-by-point response that takes into account the points raised. Please highlight all changes in the manuscript text file.

Please use the following link to submit your revised manuscript, point-by-point response to the referees' comments (which should be in a separate document to any cover letter), a tracked-changes version of the manuscript (as a PDF file) and the completed checklist:

Link Redacted

We hope to receive your revised paper within six weeks; please let us know if you aren't able to submit it within this time so that we can discuss how best to proceed. If we don't hear from you, and the revision process takes significantly longer, we may close your file. In this event, we will still be happy to reconsider your paper at a later date, as long as nothing similar has been accepted for publication at Communications Earth & Environment or published elsewhere in the meantime.

Please do not hesitate to contact us if you have any questions or would like to discuss these revisions further. We look forward to seeing the revised manuscript and thank you for the opportunity to review your work.

Best regards,

Heike Langenberg, PhD
Chief Editor
Communications Earth & Environment

On Twitter: @CommsEarth

EDITORIAL POLICIES AND FORMATTING

Editorial Policy: [Policy requirements](https://www.nature.com/documents/nr-editorial-policy-checklist.pdf) (Download the link to your computer as a PDF.)

Furthermore, please align your manuscript with our format requirements, which are summarized on the following checklist: [Communications Earth & Environment formatting checklist](https://www.nature.com/documents/commsj-phys-style-formatting-checklist-article.pdf)

and also in our style and formatting guide [Communications Earth & Environment formatting guide](https://www.nature.com/documents/commsj-phys-style-formatting-guide-accept.pdf) .

*** DATA: Communications Earth & Environment endorses the principles of the Enabling FAIR data project (<http://www.copdess.org/enabling-fair-data-project/>). We ask authors to make the data that support their conclusions available in permanent, publically accessible data repositories. (Please contact the editor if you are unable to make your data available).

All Communications Earth & Environment manuscripts must include a section titled "Data Availability" at the end of the Methods section or main text (if no Methods). More information on this policy, is available at <http://www.nature.com/authors/policies/data/data-availability-statements-data-citations.pdf>.

If a community resource is unavailable, data can be submitted to generalist repositories such as [figshare](https://figshare.com/) or [Dryad Digital Repository](http://datadryad.org/). Please provide a unique identifier for the data (for example a DOI or a permanent URL) in the data availability statement, if possible. If the repository does not provide identifiers, we encourage authors to supply the search terms that will return the data. For data that have been obtained from publically available sources, please provide a URL and the specific data product name in the data availability statement. Data with a DOI should be further cited in the methods reference section.

REVIEWER COMMENTS:

Reviewer #2 (Remarks to the Author):

Review of "Surge in future record shattering precipitation driven by increasing variability" by Iris de Vries, Sebastian Sippel, Joel Zeder, Erich Fischer and Reto Knutti

GENERAL COMMENTS

This is an excellent paper which I very much enjoyed reviewing (and I don't say that often). It is well-suited for publication in Communications Earth & Environment. It creates a very useful framework for interpreting expected changes in record breaking extreme precipitation events which I believe the climate research community has been missing. I have made some comments below which will likely require some modifications to the paper, but they are all rather minor. I hope that they help to improve the quality of the paper which is already very high.

SPECIFIC COMMENTS

My first thought regarding the paper is the use of the word 'shattering' in the title rather than the more expected word 'breaking'. Usually we refer to record breaking events rather than record shattering events. This begs the question of how much does a record need to be exceeded by for it to qualify as record 'shattering' rather than record 'breaking'. I expect/hope to see a formal definition of the difference in the manuscript. The abstract talks about records being broken 'by large margins' as being 'shattering'. So I expect to see how large that margin needs to be. Ah good, I see something to this effect around lines 44-45. Line 46 again refers to 'by a large margin' but I really would like to see 'large' quantified. Maybe just something like 'breaking the previous record by at least the standard deviation calculated over a predefined historical period'.

Papers published in Communications Earth & Environment should be accessible to quite a wide audience. The phrase 'forced with SSP3-7.0' may be a little too terse and may need to be expanded on so that non-expert readers are aware that this refers to a fairly high greenhouse gas emissions scenario.

Line 15: Here is the use again of 'record shattering' as opposed to 'record breaking' without any formal definition of what 'record shattering' means. I think you need to get that definition in place as early as possible in the paper, i.e., when are you going to use the phrase 'record breaking' and by how much does the record need to be broken for you to rather use the phrase 'record shattering'. I found out later in the paper what you mean but as long as it remains unresolved early in the paper, it will niggle at the back of the mind of the reader. Without that definition in place, sentences such as 'We show that CMIP6 models forced with SSP3-7.0 project much higher probability of record shattering extreme precipitation in a changing climate relative to a stationary climate by the end of the century' don't mean anything.

Line 63: I don't think that the increase is exponential. First, even Clausius-Clapeyron is not really exponential (it's some power law or geometric progression - there is no $\exp()$ in the Clausius-Clapeyron formula. And then there are regions which either exceed or fall short of the Clausius-Clapeyron expectation of water holding capacity of the atmosphere. Again on line 70, unless you are really sure that the response is exponential, you may want to use a less prescriptive phrase. Perhaps just something like 'highly non-linear'?

Lines 73-80: As I read this paragraph, my first thought is how much confidence do we have that the CMIP models adequately incorporate the processes underlying extreme precipitation, especially given that many of these processes occur on scales that are smaller than the prescribed spatial scale of the model and therefore need to be parameterized. I expect to see some caveats to this effect early on in the paper. I did see some later in the paper but I think that more needs to be said up front in the paper about model limitations for the purposes of such a study. I am also very much hoping that the paper will differentiate between the CMIP6 models based on their equilibrium climate sensitivity (ECS). In particular, I am keen to see whether the models with high ECS simulate record 'shattering' more frequently than models with low ECS which may, intuitively, be what is expected, but may not necessarily be the case.

Figure 1: I am finding the legend in this figure confusing. The legend shows two solid black lines, one labelled 'E-OBS' and one labelled 'Regional average'. So which is it? Then there is a yellow dashed line that isn't described at all in the legend. And also a blue dashed line that isn't described in the legend. I am very pleased to see the yellow shading around your 1-in-1000 year precipitation depth value. People often don't show uncertainties on GEV fit coefficients.

Line 146: I am interested that you kept the shape parameter constant. Did you not have enough 5-year block maxima extracted from the 100-member ensemble to derive a statistically robust non-stationary component to the shape parameter?

Line 167: Is it that the change in precipitation extremes is significantly higher in elevated regions or that it is higher on the windward side of elevated regions, i.e., where orographically forced precipitation in the presence of moist flows (e.g. atmospheric rivers) will lead to precipitation extremes.

Line 175: I think you need to say more here that just 'Arctic amplification'. I know what you mean but not all readers may readily make the connection between 'strong winter warming' and 'Arctic amplification'.

Line 188: I expected that this paper would be cited here: Sippel, S.; Mitchell, D.; Black, M.T.; Dittus, A.J.; Harrington, L.; Schaller, N. and Otto, F.E.L., Combining large model ensembles with extreme value statistics to improve attribution statements of rare events, *Weather and Climate Extremes*, doi:10.1016/j.wace.2015.06.004, 2015.

Line 194: This is a very nice and useful finding, and great to see it clearly demonstrated here.

Line 215: Just to be clear, when you say 'only by keeping σ constant' I assume you mean that having evaluated the non-stationary GEV fit (which, as you said, you do not repeat), you now consider only the stationary fit component for σ and ignore the non-stationary component? Maybe you want to detail that just a little bit more explicitly.

Line 221: There is no expectation that these two components should be linearly additive is there?

Line 223: I think that what you are concluding here is correct but I am no more than 90% certain. An easy check would be to actually fit two different GEV models. In both the shape parameter would be stationary (as you have it). In model (1) the location parameter would be expanded in a covariate and the scale parameter not. In model (2) the scale parameter would be expanded in a covariate and the location parameter not. We would then expect model (2) to have larger explanatory power than model (1), right? If I've got that right, and it's easy enough to do that test, I think it would add just a little bit more evidence to support your conclusion.

Line 238: You say 'is likely to be stronger'. Do you mean is likely to be stronger compared to 5yRx1d?

Line 238: You talk here about 'Stronger trends' and I may have lost the plot a bit but stronger trends in what exactly? I guess you mean stronger trends in precipitation - that would make sense. Or do you mean 'stronger non-stationarity' of the GEV fit coefficients? I would suggest reserving the term 'trend' strictly for cases where you mean a rate of change with time (which may well be what you mean here). But if you're talking about that relative magnitudes of the stationary and non-stationary GEV fit coefficients for one of the three fit parameters, it might be better to say 'stronger non-stationarity' (or something else that is descriptive), rather than using the word 'trend'. If the covariate that you're using to capture the non-stationarity is

unitless, then you could indeed look at the ratio of the stationary and non-stationary GEV fit coefficients (which I see you have done in some places). See, then later in the sentence, you say 'trend in μ ' whereas I think what you really mean is the non-stationarity in μ .

Line 279: True, but the boundary of the region defined by your 8 or 9 selected cells is equally arbitrary. That said, I don't disagree with you. From a policy perspective, whether an extreme event falls within one cell or another of some contiguous jurisdictional area, that some emergency management authority is responsible for, is irrelevant.

Line 293-294: I guess that when you say 'the number of record shattering events in the 21st century is higher for Pakistan and Lagos' you mean compared to the 20th century?

Figure 3: Looking at the Lagos results I noticed a couple of anomalies:

- 1) There is a red dot (record shattering event) around 1990. However, one of the other grid cells shows an even higher peak in the same year but this is not flagged as a record shattering event. How come?
- 2) Some time shortly after 2000 there are some quite high events that look higher than the 1990 event but are not flagged as record shattering.
- 3) There is a quite low event around 2040 that is flagged as record shattering that appears to have some neighbours in other years that are higher but not flagged a record shattering.

Having read all the way to the end of the paper I now better understand these apparent anomalies and could probably answer my own questions. But I suspect other readers may have the same questions in their minds at this point. By providing the equation on page 14 earlier in the document, and describing in more detail and much earlier in the paper how record 'shattering' is defined, I think would resolve those outstanding questions.

Line 322: Provides valuable insights into what exactly?

Line 325: I found this some what confusing. If I understand you correctly, I think that this would be better worded as 'The effects of non-stationarity in the mean (μ , dotted)'.

Line 328: Personally I find Figure S14 very instructive and wonder if it shouldn't be moved to the main part of the manuscript. And can you please label the panels in Figure S14 with the associated regions.

Line 345: Unless you are sure that the increase is truly exponential, I would avoid using this word.

Line 345-346: I think that some care needs to be exercised here. I am looking at equation (3) on page 13: the point here is that it is not that μ_1 changes, but rather that because $g\mu(t)$ is non-linear, $\mu(t)$ becomes non-linear. So I think that it is worth pointing out here that the second order structure in $\mu(t)$ is driven entirely by the behaviour of $g\mu(t)$. Had you chosen a different covariate for the non-stationarity, the structure in $\mu(t)$ would have been different. Isn't the structure therefore largely prescribed by your choice of covariate and not something emergent from the data? There are some interesting words to this effect on line 526 where you say 'the GEV distribution is allowed to vary in a user-defined way'. So here you do indeed point out that the behaviour is, to some extent, user-defined.

Lines 364-365: The conclusion that 'if extreme precipitation variability increases strongly, record shattering probabilities respond strongly' is almost statistically self-evident right? This conclusion could have been made (and probably was) 50 years ago - it doesn't require the analysis presented in your paper. So I think that you need to say something more nuanced here. Perhaps something about how the interplay between an underlying secular trend (described by μ and its non-stationarity) and potentially changing variability (described by σ and its non-stationarity) influences the 'emergence' of a climate signal (in your case in the form of record shattering precipitation events) from climate noise. I guess what I am saying here is that your analysis is far more valuable than what is portrayed in this paragraph. I would suggest that you don't undersell what you have done with the conclusion that 'if extreme precipitation variability increases strongly, record shattering probabilities respond strongly'.

Line 378: I am not sure that you should be referring to this as an 'exponential trend'. When I look at equation (4), that looks more like a geometric functional form; I don't see any $\exp()$ in your formula. Maybe I am nitpicking here - your call.

Line 422: I was really hoping to see something here about how record shattering is likely to depend in the ECS of the underlying model. But perhaps that will be coming in a follow-up paper? Given all of the debate and discussion about 'hot' models (i.e. models with high ECS), and, as you point out, the societal interest in changing likelihood of record shattering events, understanding the dependence of the record shattering probabilities on model ECS would gain a 'lot' of attention. I hope that's going to be your next paper - submit to Nature.

Line 433: And hopefully under different ECS.

Line 442: I am also interested in how much the conclusions you have drawn depend on the choice of non-stationarity covariate, i.e. the $g\mu(t)$ and $g\sigma(t)$ in equation (3). Could alternative, equally scientifically justifiable covariates have been selected that would have resulted in (slightly) different conclusions being drawn?

Line 449: Yes, I think that the framework that you have presented is very valuable and certainly the most valuable thing I got from the paper. That's the thing I am going to remember most from the paper. Given that, you may want to reframe your abstract a little?

Line 529: But if you have an ensemble of several thousand members (e.g. from the weather@home model database) you should be able to determine with some statistical robustness the non-stationarity in the shape parameter right? So it is not a priori uncertain. You just need way more data to get a statistically robust estimate of the non-stationarity in the shape parameter. Are there not other papers published that show that excluding non-stationarity in the shape parameter leads to the most parsimonious structure of the GEV model for most climate related time series?

Line 533: What if, instead of using the ensemble mean, you just used the annual mean global mean surface temperature anomaly? That would have been equally scientifically defensible and, in fact, many other papers have done exactly that. How different would your conclusions have been?

Line 543: Ah good. I am glad you did that test. That's very useful to know.

Line 547: And then if you really want to be ambitious, you could further expand the five fit coefficients in functions of latitude and longitude so that you can fit the GEV model to block maxima from a small domain around the cell of interest and thereby further reduce the uncertainties on the fit coefficients. For example, replace μ_0 with $\mu_{00} + \mu_{01} * d_lat + \mu_{02} * d_lon$ where d_lat is the latitude difference between the target cell and the neighbouring cell and d_lon is the longitude difference between the target cell and the neighbouring cell. Assuming a linear dependence on latitude and longitude across a grid of 5x5 cells, which I think would be entirely justified, would increase the number of block maxima by a factor of 25 while increasing the number of fit coefficients only by a factor of 3 which I think would help to reduce the uncertainties in the GEV fit coefficients. I am not suggesting that you do that for this analysis. but it might be worth keeping in mind for the future.

Line 576: I like this formula towards the bottom of page 14 that describes your determination of 'record shattering'. Maybe it should be brought up into the main body of the manuscript and referred to quite early in the paper since understanding of what you mean by 'record shattering' is essential to understanding the whole paper.

SUGGESTED TYPOGRAPHICAL AND GRAMMATICAL CORRECTIONS

The suggestions below are very much just suggestions and the authors should feel free to incorporate them, or not, as they see fit.

Line 33: Replace 'degree in which' with 'degree to which'.

Line 42: Replace 'trends or records' with 'trends or records being broken' - the records still exist, they're just not being broken.

Line 62: Replace 'also these will occur' with 'these will also occur'.

Line 105: Replace 'in line with' with 'consistent with'.

Line 117: Replace 'simulated data' with 'model simulations'.

Line 159: Replace 'are of import' with 'are important' - unless you want to sound fancy.

Line 167: Replace 'change of precipitation' with 'change in precipitation'.

Line 175: Replace 'arctic' with 'Arctic'.

Line 183: Replace 'is very similar as well' with 'is also very similar'.

Line 184: I think that it would be better to refer to these as 'zonal means' rather than 'latitudinal averages'.

Line 200: Replace 'northern hemispheric' with 'NH'. You've defined the acronym so you might as well use it.

Line 274: Replace 'would compound spatial extent' with 'would compound the spatial extent'.

Line 324: Replace 'only the σ -trend is' with 'only the σ non-stationarity is'.

Line 326: Replace 'aspects' with 'behaviour'.

Line 334: Replace 'the μ -trend only does not increase record shattering probability, yet, it does manifest as a lessening of the decreasing probability' with 'not only does the μ -trend increase record shattering probability, it also manifests as a reduction of the decreasing probability'. Although the phrase 'lessening/reduction of the decreasing probability' I still find confusing. You may need to give this some further thought.

Line 479: Replace 'data was' with 'data were'. And likewise elsewhere. The word 'data' is plural (like the word 'values'). Its singular form is datum (value).

Lien 546: Replace 'the amount of estimated parameters' with 'the number of estimated parameters'.

Reviewer #3 (Remarks to the Author):

The manuscript uses extreme value statistics and CMIP6 models (with a focus on CESM2-LE) to show there is a higher probability of record shattering precipitation driven by increasing variability by the end of the century. This has implications for developing adaptation policies and mitigation strategies related to anticipating to rare events. While the focus of the manuscript is on statistics of precipitation, the authors also show physical 'realness' of their results, at very detailed levels (examples of conditions (IVT, SLP, updraft anomalies) during "record breakers" in the observational and modeled simulated results) as well as explanation of results over large geographic areas. The manuscript concludes with an analytical rationale to explain the patterns being seen. Overall, it provides compelling evidence for the importance of changing variability of precipitation, rather than just mean, which has also been highlighted in prior studies. At the same time, I have some concerns related to the strength of the results for one CMIP6 model only and the approach taken, both in the non-stationary extreme value statistics and in the comparison of the modeled to observational data, which should be further explained.

General Comments:

One of my main concerns is that the strength of the "record shattering precipitation driven by increasing variability" is mainly true for one model only. While this conclusion is transparent in the figures and throughout the manuscript, it seems as if the importance of this should be more upfront (in the title or abstract). For example, there are some snippets of text that seem to be hidden away in the manuscript and in the supplemental section that basically say CESM2-LE data shows the most influence from scale parameter of any model, and hence variability, so the results are not robust across other future projections. BUT the conclusion holds that IF extreme precipitation variability increases strongly, record shattering probabilities respond strongly. If this is the case, there should be more specific emphasis on the single model aspect of these findings in the title and the abstract, as those are absolute in saying that shattering precipitation is driven by increasing variability in the future, across CMIP6 models.

Following on, I was surprised that there was no discussion of bias correction of the GCM precipitation data. The main point of the paper is to evaluate the changing statistics of very rare events, so there needs to be a description/quantification of how the statistics of the simulated historical precipitation data preserve the statistics of the observed precipitation data over the same period of time. If the historical simulations don't reproduce the statistics of the historical observed data, then the change values across to the future can't necessarily be trusted. A qualitative comparison of the deviation around the anomalies isn't enough – the variation in the distribution – especially at the tails – needs to be quantified. Some statistical bias correction methods commonly used are linear scaling, quantile mapping, distribution mapping, and cumulative distribution function transforms. Other methods have combine approaches (developed hybrid methods) for preserving the extreme and whole ends of the distribution. Either way, if the data doesn't need to be bias corrected to retain the statistics of the distribution, this needs to be clearly shown.

What validation tests did the authors do to ensure the significance and goodness of fit of their nonstationary GEV models? When fitting models with several covariates, it's best practice to ensure parsimony of the best model. This is often done with a combination of diagnostic plots, information criteria, and likelihood ratio tests. I suggest the authors test and show the significance of their model, with non-stationarity in the location and scale parameter compared to a stationary model fit as well as a comparison of a model with just nonstationarity in the scale parameter OR nonstationarity in the shape parameter to their model of nonstationary in the location and scale parameter.

Specific Comments/Line-by-Line:

Related to the nonstationary GEV models – I'm also wondering why covariates related to the ensemble mean and ensemble variability were used instead of letting them take the form $\mu(t) = \mu_0 + \mu_1(t)$ (or some other nonlinear trend), letting the μ_1 (or σ_1) scale by the extreme data changing over time, rather than an external covariate?

Where does CESM2-LE fit in the climate models in terms of precipitation? The authors reveal there are similar patterns to other climate models, but different projections of precipitation distribution changes. In general, what does the literature say about the variation in precipitation across the models?

Do the patterns of shattering probabilities, relate at all to the shape parameter of the model? I'd assume there is also geographic variability of the shape parameter, usually heavier tailed around the tropics. Similarly, does the shape parameter alter the analytical results (i.e. would you see the same patterns and trends regardless of heavy, light, or bounded tail)? What is the shape parameter for the analytical results in section 3 of the paper?

What are the limitations of the work related to resolution /modeling of tropical cyclones in the data set? How well are tropical cyclones represented in the extreme precipitation data here?

Line 606 – 607 says “it is important to view trend-separation in the GEV context in terms of location and scale” (rather than mean and standard deviation). However the section starting on line 310, is titled “The role of mean and variability for regional changes...” and the authors discuss mean and variability, and specifically call “mean trend only” (μ , dotted), and small mean trends and small variability trends, rather than using the location terminology. This happens also in lines 355 – 357 and in Section 3, analytical rationale which also discusses different mean and variability trend combinations.

How do the authors get the confidence intervals on 500 and 5000 year events in Figure 3? They seem way too constrained to represent the uncertainty in the GEV model fit.

Communications Earth & Environment is committed to improving transparency in authorship. As part of our efforts in this direction, we are now requesting that all authors identified as ‘corresponding author’ create and link their Open Researcher and Contributor Identifier (ORCID) with their account on the Manuscript Tracking System prior to acceptance. ORCID helps the scientific community achieve unambiguous attribution of all scholarly contributions. You can create and link your ORCID from the home page of the Manuscript Tracking System by clicking on ‘Modify my Springer Nature account’ and following the instructions in the link below. Please also inform all co-authors that they can add their ORCID to their accounts and that they must do so prior to acceptance.

Author Rebuttal letter: The author’s response to these comments can be found at the end of this file.

Version 1:

Decision Letter:

Dear Ms de Vries,

Your manuscript titled "Increasing extreme precipitation variability plays key role in future record-shattering event probability" has now been seen by our reviewers, whose comments appear below. In light of their advice we are delighted to say that we are happy, in principle, to publish a suitably revised version in Communications Earth & Environment under the open access CC BY license (Creative Commons Attribution v4.0 International License).

We therefore invite you to revise your paper one last time to address the remaining concerns of our reviewers. At the same time we ask that you edit your manuscript to comply with our format requirements and to maximise the accessibility and therefore the impact of your work.

EDITORIAL REQUESTS:

****Please take care to match our formatting and policy requirements. We will check revised manuscript and return manuscripts that do not comply. Such requests will lead to delays. ****

SUBMISSION INFORMATION:

OPEN ACCESS:

Communications Earth & Environment is a fully open access journal. Articles are made freely accessible on publication under a [CC BY license](http://creativecommons.org/licenses/by/4.0) (Creative Commons Attribution 4.0 International License). This license allows maximum dissemination and re-use of open access materials and is preferred by many research funding bodies.

For further information about article processing charges, open access funding, and advice and support from Nature Research, please visit <https://www.nature.com/commsenv/article-processing-charges>

At acceptance, you will be provided with instructions for completing this CC BY license on behalf of all authors. This grants us the necessary permissions to publish your paper. Additionally, you will be asked to declare that all required third party permissions have been obtained, and to provide billing information in order to pay the article-processing charge (APC).

Link Redacted

Best regards,

Heike Langenberg, PhD
Chief Editor
Communications Earth & Environment

On X(Twitter): @CommsEarth

REVIEWERS' COMMENTS:

Reviewer #2 (Remarks to the Author):

Second review of "Surge in future record shattering precipitation driven by increasing variability" by Iris de Vries, Sebastian Sippel, Joel Zeder, Erich Fischer and Reto Knutti

The authors have done an excellent job in responding to my first review and the paper is now much improved and ready to be published. I think that if the review and the authors response are provided along with the paper this will further add to the quality of the manuscript as a whole. Great paper. Time for it to be published. I look forward to reading the published version.

Reviewer #3 (Remarks to the Author):

I am satisfied with the authors' detailed responses and additional analyses. I think this paper is a valuable contribution to the literature. Well done!

Author Rebuttal letter: The author's response to these comments can be found at the end of this file.

Dear reviewers,

We thank you very much for your thoughtful and helpful reviews.

Below we inserted our replies in **black** to your comments in **grey**. We quote the added/changed text where relevant, and added line numbers for the revised manuscript of the corresponding sections. Besides the changes requested by the reviewers, some additional changes were made to meet the style guide (e.g. to figure captions and the abstract), as well as some aesthetic changes. We also attach a track-changes version of the revised manuscript, in which all changes are clearly marked.*

We hope our replies and changes are to your satisfaction.

Best,

Iris de Vries, on behalf of all authors

* Throughout the manuscript, we added a hyphen "-" to record-shattering whenever the word is used as an adjective (i.e. record-shattering event). The hyphens are left out of the track-changes file for readability.

Changes made independently of reviewers

1. Abstract shortened to fit the style guide
2. Last introductory paragraph edited to fit the style guide (major results in the last paragraph).
3. Changed heading to "Results **and Discussion**" to fit the style guide
4. Removed all quotation marks (apart from those referring to a line in a figure) to fit the style guide.
5. Removed "not shown" results to fit the style guide.
6. Changed figure captions to include a brief title to fit the style guide.
7. Implemented a number of aesthetic textual changes, all documented in the latex-diff file. Sometimes larger structural changes were made (e.g. re-ordering of paragraphs in conclusion), but no content was changed other than those content additions requested by reviewers.

Point by point response to comments

Reviewer #2 (Remarks to the Author):

Review of "Surge in future record shattering precipitation driven by increasing variability" by Iris de Vries, Sebastian Sippel, Joel Zeder, Erich Fischer and Reto Knutti

GENERAL COMMENTS

This is an excellent paper which I very much enjoyed reviewing (and I don't say that often). It is well-suited for publication in Communications Earth & Environment. It creates a very useful framework for interpreting expected changes in record breaking extreme precipitation events which I believe the climate research community has been missing. I have made some comments below which will likely require some modifications to the paper, but they are all rather minor. I hope that they help to improve the quality of the paper which is already very high.

We thank the reviewer for these very kind words!

SPECIFIC COMMENTS

My first thought regarding the paper is the use of the word 'shattering' in the title rather than the more expected word 'breaking'. Usually we refer to record breaking events rather than record shattering events. This begs the question of how much does a record need to be exceeded by for it to qualify as record 'shattering' rather than record 'breaking'. I expect/hope to see a formal definition of the difference in the manuscript. The abstract talks about records being broken 'by large margins' as being 'shattering'. So I expect to see how large that margin needs to be. Ah good, I see something to this effect around lines 44-45. Line 46 again refers to 'by a large margin' but I really would like to see 'large' quantified.

Maybe just something like 'breaking the previous record by at least the standard deviation calculated over a predefined historical period'.

We fully agree, the definition of record-shattering came too late. We have now specified in L43 what the "large margin" is:

"In this work, record-shattering events are defined as events that break, indeed shatter, the historical maximum value (record) by a margin of at least one pre-industrial standard deviation, defined as the mean standard deviation in the period 1850-1949."

And also refer already to the margin in the abstract:

"we estimate the probability of precipitation events that shatter records by a margin of at least one pre-industrial standard deviation in climate models" (L13)

Papers published in Communications Earth & Environment should be accessible to quite a wide audience. The phrase 'forced with SSP3-7.0' may be a little too terse and may need to be expanded on so that non-expert readers are aware that this refers to a fairly high greenhouse gas emissions scenario.

We have removed SSP3-7.0 from the abstract. In the introduction, we do mention SSP3-7.0, mention it is a high warming scenario, and refer to the relevant literature. In L240, L350, and L482 (conclusions) we emphasise again that SSP3-7.0 is a high warming scenario.

Line 15: Here is the use again of 'record shattering' as opposed to 'record breaking' without any formal definition of what 'record shattering' means. I think you need to get that definition in place as early as possible in the paper, i.e., when are you going to use the phrase 'record breaking' and by how much does the record need to be broken for you to rather use the phrase 'record shattering'. I found out later in the paper what you mean but as long as it remains unresolved early in the paper, it will niggle at the back of the mind of the reader. Without that definition in place, sentences such as 'We show that CMIP6 models forced with SSP3-7.0 project much higher probability of record shattering extreme precipitation in a changing climate relative to a stationary climate by the end of the century' don't mean anything.

Fixed, see changes referred to above.

Line 63: I don't think that the increase is exponential. First, even Clausius-Clapeyron is not really exponential (it's some power law or geometric progression - there is no $\exp()$ in the Clausius-Clapeyron formula. And then there are regions which either exceed or fall short of the Clausius-Clapeyron expectation of water holding capacity of the atmosphere. Again on line 70, unless you are really sure that the response is exponential, you may want to use a less prescriptive phrase. Perhaps just something like 'highly non-linear'?

We'd like to provide a general reply here that applies to all the comments below on the exponential relationship between temperature and precipitation, since this issue comes up a few times:

The confusion might come from the fact that any variant of the base exponential function $y = \exp(x)$, modified with some scalar coefficient a : $y = \exp(ax)$ can be written as $y =$

$\exp(a)^x$ where $\exp(a)$ is a constant. The $\exp()$ then disappears from the formula, but the relationship between y and x is still exponential (and thus y has a steepening trend with linear x , which is essential for record breaking and shattering).

We agree with the reviewer that the solution of the Clausius Clapeyron equation is not exactly exponential under all possible circumstances, for instance because other terms in the equation may depend indirectly on temperature as well (such as the specific latent heat of evaporation). However, in almost all practical settings relevant for climate/meteorological applications, the Magnus formula gives a very good approximation as a (simplified) version of the Clausius Clapeyron equation for lower tropospheric temperatures:

$$e_s \approx 6.1 \cdot \exp\left(\frac{17.6 \cdot T}{T+243}\right) \approx 6.1 \cdot 1.07^T$$

Where e_s is saturation vapour pressure in hPa and T is temperature in degrees Celsius. Given that T is in degrees Celsius, the denominator is only weakly dependent on temperature. This results in an approximate increase of 7% per K, reflecting the approximately exponential increase in atmospheric water vapour holding capacity with warming.

So even though this is an approximation of the Clausius Clapeyron equation, it is the standard approximation used in the field. At the instance referred to here, L62, we state the increase is "approximately exponential". At later instances in the paper, where comments below refer to, we have refrained from stating the approximateness explicitly since we believe the intended audience is familiar with the common approximation of water vapour pressure scaling with a certain percentage change per degree of warming.

We note that the 7% value does not hold everywhere, which we also say in L66: "increases in precipitation intensity scale with event rareness". The importance, however, of the exponential increase is that it implies an increasing trend slope, which comes back later in the analytical rationale: an increasing trend slope implies increasing record breaking probabilities (whereas linear (constant) trend slopes do not).

Lastly, we have made the statement in L62-65 more nuanced by outlining that the 7% water vapour increase does not translate directly to precipitation, and cite two sources (that were cited multiple times at other locations in the paper) that explicitly address precipitation scaling with temperature:

"Albeit this does not directly translate to the change in extreme precipitation with warming – precipitation efficiency and dynamical processes modify the scaling rate –, exponential scaling of extreme precipitation with local temperature has been shown to hold in models and observations, especially in mid- to high latitudes."

Lines 73-80: As I read this paragraph, my first thought is how much confidence do we have that the CMIP models adequately incorporate the processes underlying extreme precipitation, especially given that many of these processes occur on scales that are smaller than the prescribed spatial scale of the model and therefore need to be parameterized. I expect to see some caveats to this effect early on in the paper. I did see some later in the paper but I think that more needs to be said up front in the paper about model limitations for the purposes of such a study. I am also very much hoping that the paper will

differentiate between the CMIP6 models based on their equilibrium climate sensitivity (ECS). In particular, I am keen to see whether the models with high ECS simulate record 'shattering' more frequently than models with low ECS which may, intuitively, be what is expected, but may not necessarily be the case.

We agree that model resolution and intermodel differences greatly affect the results. We also agree that we have to make this point more clear from the start, and added at the end of the introduction, L87-94:

"[...] precipitation record-breaking and -shattering behaviour differs between climate models and between simulations and the real world due to, among other things, uncertainty in future emissions, climate sensitivity, and forced dynamical changes, small scale processes that are not resolved in models, and natural variability. While acknowledging these uncertainties and the limitations of climate model simulations, we identify a framework for the analysis of changing record-shattering precipitation probability."

We show different models in Fig 2 and mention that model differences including climate sensitivity affect the results, and we mention this again in the conclusions. Also Supplementary Section 6 contains some more across-model assessment. We hope this conveys that we want to be fully transparent about the sensitivity of our results to ECS and model differences, which should be taken into account in the interpretation.

We fully agree with the reviewer that the influence of ECS (or TCR) on record breaking or shattering precipitation is a very relevant and interesting topic. We deliberately chose, however, not to include a systematic analysis on it in this work, since we want to keep the scope small and focused on how distributional characteristics that generally apply to precipitation (high variability applies regardless of climate sensitivity) translate to record shattering probabilities.

Figure 1: I am finding the legend in this figure confusing. The legend shows two solid black lines, one labelled 'E-OBS' and one labelled 'Regional average'. So which is it? Then there is a yellow dashed line that isn't described at all in the legend. And also a blue dashed line that isn't described in the legend. I am very pleased to see the yellow shading around your 1-in-1000 year precipitation depth value. People often don't show uncertainties on GEV fit coefficients.

Thanks for pointing this out. One legend is the linetype legend (CESM2-LE versus E-OBS) and the other legend is the colour legend: i.e. the colours hold for both linetypes. In order to make this clear we now simply added these titles to the legend ("Linetype" and "Colour") to avoid confusion. The same change was made in Fig. 2 and 4.

Line 146: I am interested that you kept the shape parameter constant. Did you not have enough 5-year block maxima extracted from the 100-member ensemble to derive a statistically robust non-stationary component to the shape parameter?

Even for large sample sizes, it is notoriously difficult to estimate the shape parameter, as we say in L580 and show also in Supplementary Sect. 2 and Supplementary Fig. S17. See Coles

(2001) for more on GEV considerations. In climate applications, it is common practice to keep the shape parameter constant in a non-stationary setting. An additional reason to do so, is to keep the distributional changes tractable. Furthermore, there is no overwhelming evidence that the shape parameter of extreme precipitation distributions would change with climate change, nor how. Given the high uncertainty of the shape parameter, it is most conservative to keep it constant. We have now added a very short reasoning to (former) L146, and refer to the method section for further details:

L168: " ξ is kept constant with time for reasons of accuracy and tractability, see Sect. 4."

See also our reply to the last general comment of reviewer 3; we have added Supplementary Sect. S2, which validates the GEV setup we use.

Just for illustration, we show below the difference between GEV parameters resulting from the non-stationary fit (as we use in the study) and GEV parameters from fits made for each 5-year period separately for the three regions shown in the paper, season MJJAS. The difference between the 5-yearly parameter and the nonstationary parameter is shown as a percentage of the time-averaged value of the parameter in question. Note that the shape parameter is shown on the right y-axis in absolute terms, as the shape parameter roughly varies between -0.5 and 0.5, and percentages explode if the shape parameter goes to 0. This figure indicates that there is no systematic trend in the shape parameter. This also holds for individual grid cells and NDJFM. All of the reasons above explain why we keep the shape parameter constant.

Fig. R1 difference between GEV parameters obtained from 5-yearly GEV-fits without prescribed covariates and GEV parameters obtained from a non-stationary fit with prescribed covariates for μ and σ .

Line 167: Is it that the change in precipitation extremes is significantly higher in elevated regions or that it is higher on the windward side of elevated regions, i.e., where orographically forced precipitation in the presence of moist flows (e.g. atmospheric rivers) will lead to precipitation extremes.

The resolution of the models we use is not high enough to resolve this difference, and this holds for the data used in the reference we cite (Ombadi et al. (2023)) as well (horizontal resolution ranges from 60-100km). The reasons for high precipitation in mountain regions and a strong record-shattering response are complex - both the thermodynamic response with elevation and the orographic (convective) lift causing extreme precipitation events are

important here. The study we cite attributes a large part of the higher increase in rainfall in mountainous regions to a transition from snow to rain (i.e. no increase in total precipitation (solid and liquid) per se).

Thanks for pointing us to this remark, which we have specified now (L193):

"Also mountain ranges, where both thermodynamic and dynamic effects are at play, stand out. This is concerning from a flood-risk perspective, especially in combination with the recent finding that snow-to-rain transition amplifies the increase in rainfall in elevated regions.

Line 175: I think you need to say more here than just 'Arctic amplification'. I know what you mean but not all readers may readily make the connection between 'strong winter warming' and 'Arctic amplification'.

We removed this clause and now simply refer to strong projected NH warming, with a reference to the IPCC WG1 report chapter 4 that shows this.

Line 188: I expected that this paper would be cited here: Sippel, S.; Mitchell, D.; Black, M.T.; Dittus, A.J.; Harrington, L.; Schaller, N. and Otto, F.E.L., Combining large model ensembles with extreme value statistics to improve attribution statements of rare events, *Weather and Climate Extremes*, doi:10.1016/j.wace.2015.06.004, 2015.

Thanks, this is a relevant reference indeed, we've added it.

Line 194: This is a very nice and useful finding, and great to see it clearly demonstrated here.

Thanks!

Line 215: Just to be clear, when you say 'only by keeping σ constant' I assume you mean that having evaluated the non-stationary GEV fit (which, as you said, you do not repeat), you now consider only the stationary fit component for σ and ignore the non-stationary component? Maybe you want to detail that just a little bit more explicitly.

We keep μ and σ constant in turn by setting them to their 1850-1949 mean value of the nonstationary fit. We explain this in the method section, that we refer to in the line in question. We added the sentence "We keep the parameters constant by setting their value to the 1850-1949 mean of the non-stationary parameter fit at all time steps." in L251 to already give the essential information here.

Line 221: There is no expectation that these two components should be linearly additive is there?

No they are not. We failed to mention that, but it is in fact important to flag that, so we've added it now in L254-255.

Line 223: I think that what you are concluding here is correct but I am no more than 90% certain. An easy check would be to actually fit two different GEV models. In both the shape parameter would be stationary (as you have it). In model (1) the location parameter would be expanded in a covariate and the scale parameter not. In model (2) the scale parameter would be expanded in a covariate and the location parameter not. We would then expect model (2) to have larger explanatory power than model (1), right? If I've got that right, and it's easy enough to do that test, I think it would add just a little bit more evidence to support your conclusion.

We like the reviewer's suggestion, but we think this approach would not directly address the research question posed here, for the following reason: the goal of the separation is to keep one GEV-parameter fixed at its preindustrial value, while letting the other one undergo climate change. If we, however, do the GEV fitting procedure as the reviewer suggests, in model (1) the constant scale parameter would be optimised to make the fit best across all sample years, i.e. it would be some middle ground between 1850 and 2100, and would thus still include climate change effects, not representing a stationary pre-industrial climate. The same holds for model (2) – the fit location parameter would not have its preindustrial value.

An alternative way to approach this, would be to non-parametrically detrend the data by removing the ensemble mean (note this is not the location parameter), and re-fit a model where only the scale parameter is allowed to vary. This is, however, not consistent with the GEV-framework, since the mean of a GEV-distributed variable is dependent on the scale parameter. Removing the non-parametric mean would therefore affect the scale of the residuals as well, and not correspond to removing the effect of a trend in the location parameter. In addition, fitting a GEV with effects of nonstationarity in the location parameter only, would require intervening non-parametrically on the data to remove the effect of the scale parameter. It is not at all straightforward to remove the influence of a changing scale parameter from the data in a way that is statistically and physically sound.

Below we show in a dummy example based on synthetic data that setting the location trend to its stationary historical mean (what we do) is effectively the same as removing the fitted location trend and refitting a GEV distribution (compare red and green), whereas removing the mean leads to changes in location and scale (compare red and blue). This validates that our approach most cleanly separates the location and scale effects in a GEV context.

Fig R2: Separating effects of location and scale. We generated an artificial GEV-distributed sample (100 values times 251 years) with prescribed location and scale trend, and added normally distributed noise. We then fitted non-stationary GEVs to this sample with the smoothed annual sample mean and SD as covariates (as in the study). The resulting parameters are plotted in red. In green, we refitted a GEV to the data from which we have subtracted the red location fit. We end up with the same scale fit, and a constant location parameter of 0, as intended. In blue, we refitted a GEV to the data after removing the mean (and do not allow non-stationarity in the location parameter anymore). This leads to a different scale parameter and a nonzero location parameter.

The figure above shows that setting the location parameter to its constant historical mean while keeping the scale parameter of the original fit, is equivalent to removing the location trend and refitting a GEV in the GEV setting. This might be obvious, but we hope the test in the figure above provides some reassurance. We added a sentence to this effect in L247:

“This is consistent in a GEV framework: subtracting the fitted μ -timeseries from the data and refitting GEV distributions would result in the same σ -values.”

Separation of non-parametric mean and variance has been done before for threshold exceedance by assuming linear additiveness and subtracting the mean from the sample. The resulting probability increase is then attributed to variability changes, and this can be subtracted from the total probability increase to obtain the residual change, attributable to mean changes, see e.g. Van der Wiel & Bintanja (2021) and Wood (2023). This, however, requires the relatively strong assumption that changes in mean and variability affect probability changes in a linearly additive way, and is not generalisable to the GEV framework because the location parameter cannot be determined in a non-parametric way.

Given that the mean and the variance of GEV-distributed variables are both dependent on the location and scale parameter, it is hard to cleanly separate these things. We therefore

prefer to stay in the GEV-setting and focus on keeping location and scale constant, since this is most tractable.

Line 238: You say 'is likely to be stronger'. Do you mean is likely to be stronger compared to 5yRx1d?

Yes, we've added this now, L285. Thanks for pointing out it was unclear.

Line 238: You talk here about 'Stronger trends' and I may have lost the plot a bit but stronger trends in what exactly? I guess you mean stronger trends in precipitation - that would make sense. Or do you mean 'stronger non-stationarity' of the GEV fit coefficients? I would suggest reserving the term 'trend' strictly for cases where you mean a rate of change with time (which may well be what you mean here). But if you're talking about that relative magnitudes of the stationary and non-stationary GEV fit coefficients for one of the three fit parameters, it might be better to say 'stronger non-stationarity' (or something else that is descriptive), rather than using the word 'trend'. If the covariate that you're using to capture the non-stationarity is unitless, then you could indeed look at the ratio of the stationary and non-stationary GEV fit coefficients (which I see you have done in some places). See, then later in the sentence, you say 'trend in μ ' whereas I think what you really mean is the non-stationarity in μ .

One 'mu-' was missing: we are talking about stronger mu-trends in this section, which means indeed a stronger rate of change of mu with time. This stronger mu-trend can be achieved through a stronger coefficient on the non-stationary covariate, or as another covariate → as we derive our covariates from the variable we are fitting GEVs to, we could derive a 10yRx1d-specific covariate which would have a different (steeper) slope increase with time. Alternatively, we could keep a general covariate and scale it up with a higher coefficient. The latter would most likely introduce bias though, because there is nonlinearity in the covariate that would look different for 5yRx1d versus 10yRx1d. This is all very detailed and, we think, not essential for the point we want to convey.

The point we are making here, is that a more extreme subsample of the extreme precipitation distribution (10y vs 5yRx1d) has stronger mu-trends due to the skewed response of extreme precipitation to climate change, i.e. steeper mu-slopes with time, and lower variability because it is a biased subsample. That leads to a relatively stronger effect of the trend in mu relative to the trend in sigma. We think that the text is clear enough to convey that (now that we added the missing "mu-").

Line 279: True, but the boundary of the region defined by your 8 or 9 selected cells is equally arbitrary. That said, I don't disagree with you. From a policy perspective, whether an extreme event falls within one cell or another of some contiguous jurisdictional area, that some emergency management authority is responsible for, is irrelevant.

It is arbitrary indeed, yet more conservative in the sense that we will see more events if we consider more than one grid cell, and arguably more representative of spatial dimensions of

regions for which one local/national government would act and where an event in one corner would still have consequences for daily life on the other end of the region (traffic, shelter, financial).

Line 293-294: I guess that when you say 'the number of record shattering events in the 21st century is higher for Pakistan and Lagos' you mean compared to the 20th century?

Compared to BNLG. We've added this now, L340.

Figure 3: Looking at the Lagos results I noticed a couple of anomalies:

1) There is a red dot (record shattering event) around 1990. However, one of the other grid cells shows an even higher peak in the same year but this is not flagged as a record shattering event. How come?

Because record shattering is path dependent: for each member it is determined relative to its own records up to the date of interest. The member without the record in 1990, thus has another event in its own history that is more extreme or within one standard deviation of the 1990 event. We only show the timeseries from 1950 onwards due to the negative power relationship of record shattering with time: for the early years records shatter often due to the short time series, but we are only interested in shattering events after "spin up". Some high entries from 1950 onwards might not be record-shattering because of events that happened before 1950 in the respective grid cell and member, see below for the Lagos MJJAS view for the whole time series from 1850 onwards. See reply below where we outline textual changes to make this more clear.

Fig R3: full record shattering timeseries for Lagos MJJAS

2) Some time shortly after 2000 there are some quite high events that look higher than the 1990 event but are not flagged as record shattering.

Same reason as in 1).

3) There is a quite low event around 2040 that is flagged as record shattering that appears to have some neighbours in other years that are higher but not flagged a record shattering. Same reason as above but the other way around: this member happens to have had no events that were more extreme in its own history.

Having read all the way to the end of the paper I now better understand these apparent anomalies and could probably answer my own questions. But I suspect other readers may have the same questions in their minds at this point. By providing the equation on page 14 earlier in the document, and describing in more detail and much earlier in the paper how record 'shattering' is defined, I think would resolve those outstanding questions.

In L142-146 we modified the text to say:

"The 5yRx1d event of member i at time t is considered record-shattering if it exceeds the maximum 5yRx1d value in ensemble member i since $t=1$, i.e. the year 1850, by a margin larger than the record-shattering threshold c_{rs} . This means that a given value can constitute a record-shattering event in one ensemble member but not in another, due to the different history of the members."

In combination with the edits in the introduction, defining the record-shattering event class earlier, we think we have now defined our approach to classify record shattering events clearly. We like the reviewer's suggestion to also include the expression of (former) page 14 already here, so we have added it. Hopefully this makes it more prominent and increases chances that future readers register the information and remember it by the time they reach the regional pooling subsection.

Line 322: Provides valuable insights into what exactly?

The second part of the sentence:

"as it corroborates the effect of increasing variability on increasing record-shattering precipitation probabilities on a regional scale."

We rephrased a bit (L368) and think this is clear now.

Line 325: I found this some what confusing. If I understand you correctly, I think that this would be better worded as 'The effects of non-stationarity in the mean (μ , dotted)'

We changed it to " μ " instead of "mean" trend to be more consistent with the figure and the comparison to the sigma-trend, but other than that we do not think this requires rephrasing.

Line 328: Personally I find Figure S14 very instructive and wonder if it shouldn't be moved to the main part of the manuscript. And can you please label the panels in Figure S14 with the associated regions.

We are grateful for this suggestion. Initially we moved the additional panels to the supplementary information to reduce the amount of metrics and details we expose the reader to, but your remark encouraged us to take it up in the main text again. We adjusted the text accordingly.

Line 345: Unless you are sure that the increase is truly exponential, I would avoid using this word.

There is broad consensus, stemming from agreement of theory, models and observations, that extreme precipitation increases with temperature with a constant percentage of its current (i.e. updated) value per degree of warming. This is, by definition, an exponential increase of precipitation with warming (e.g. $P \times 1.07^T$ for a 7% increase). We've added "approximately" the first time we mention it in the introduction, see also replies to earlier comments. We think it is important to mention it explicitly here, since, as mentioned before, an exponential increase means a steepening trend with time, and the steepening mu-trend is important for record-shattering probability.

Line 345-346: I think that some care needs to be exercised here. I am looking at equation (3) on page 13: the point here is that it is not that μ changes, but rather that because $g\mu(t)$ is non-linear, $\mu(t)$ becomes non-linear. So I think that it is worth pointing out here that the second order structure in $\mu(t)$ is driven entirely by the behaviour of $g\mu(t)$. Had you chosen a different covariate for the non-stationarity, the structure in $\mu(t)$ would have been different. Isn't the structure therefore largely prescribed by your choice of covariate and not something emergent from the data? There are some interesting words to this effect on line 526 where you say 'the GEV distribution is allowed to vary in a user-defined way'. So here you do indeed point out that the behaviour is, to some extent, user-defined.

Oh absolutely! Note, however, that the covariate $g\mu(t)$ in fact is emergent from the data, since it is the ensemble mean 5yRx1d time series. In the original manuscript we already outlined what the (user-defined) covariates are, and why we chose these covariates (L163-168):

"Non-stationarity in our application implies that μ and σ vary with time as linear functions of their covariates; we here use the smoothed 5yRx1d ensemble mean as covariate for μ , and the smoothed 5yRx1d ensemble standard deviation as covariate for σ . These covariates were chosen so as to reflect the response of the 5yRx1d mean and variability to external greenhouse gas forcing in the corresponding GEV parameters. The grid cell specific trends in μ and σ can thus be considered measures of the local forced response in mean and variability of 5yRx1d."

Regarding this comment, it is true that the mu-trend will be non-linear if the covariate is non-linear. If we would have prescribed some synthetic linear or non-linear trend, this would have strongly shaped the results, regardless of what mean trend shape the data actually exhibited. Given, however, that in our case the covariate is non-linear *because* the mean 5yRx1d trend in the data is non-linear, we are confident that this statement is justified.

Lines 364-365: The conclusion that 'if extreme precipitation variability increases strongly, record shattering probabilities respond strongly' is almost statistically self-evident right? This conclusion could have been made (and probably was) 50 years ago - it doesn't require the analysis presented in your paper. So I think that you need to say something more nuanced

here. Perhaps something about how the interplay between an underlying secular trend (described by μ and its non-stationarity) and potentially changing variability (described by σ and its non-stationarity) influences the *emergence* of a climate signal (in your case in the form of record shattering precipitation events) from climate noise. I guess what I am saying here is that your analysis is far more valuable than what is portrayed in this paragraph. I would suggest that you don't undersell what you have done with the conclusion that '*if* extreme precipitation variability increases strongly, record shattering probabilities respond strongly'.

That's kind, and we see your point. We have modified the sentence to provide a bit more detail as to the non-obviousness of the finding, L411-413:

"Nonetheless, we can conclude that the distributional characteristics of extreme precipitation, in particular its high variability and skewed response to warming, result in a high sensitivity of the probability and intensity of unseen events to changes in variability."

We point out also that this sentence is not the final conclusion of the study, in the conclusion section on L491-499 we further detail what is primarily important about this finding in a broader context.

Line 378: I am not sure that you should be referring to this as an 'exponential trend'. When I look at equation (4), that looks more like a geometric functional form; I don't see any $\exp()$ in your formula. Maybe I am nitpicking here - your call.

See answers above regarding the exponential increase of vapour pressure with warming. An exponential increase with x is an increase with a constant multiplication factor for every step in x . I.e. $\mu_{\text{exp}} = \mu_0 * 1.1^{\text{dT}}$. dT is the exponent, making it exponential. It is the same as $\mu_{\text{exp}} = \mu_0 * \exp(\ln(1.1) * \text{dT})$. Given that the response of extreme precipitation to warming is commonly expressed as a percentage increase for every degree of warming, we think the 1.1 multiplication factor with dT as the exponent is more intuitive.

Line 422: I was really hoping to see something here about how record shattering is likely to depend in the ECS of the underlying model. But perhaps that will be coming in a follow-up paper? Given all of the debate and discussion about 'hot' models (i.e. models with high ECS), and, as you point out, the societal interest in changing likelihood of record shattering events, understanding the dependence of the record shattering probabilities on model ECS would gain a *lot* of attention. I hope that's going to be your next paper - submit to Nature.

Well that's very encouraging, thanks! As the rate of warming rather than the equilibrium level is governing the record shattering probability changes, one might find that high ECS models also show high warming rates (high TCR) and thus strong record shattering responses. It would be interesting to see how this propagates into μ and scale contributions for different models.

Line 433: And hopefully under different ECS.

We added "across models" (L480) to refer to differences in models such as ECS, but also other differences are likely to affect record shattering behaviour.

Line 442: I am also interested in how much the conclusions you have drawn depend on the choice of non-stationarity co-variate, i.e. the $\mu(t)$ and $\sigma(t)$ in equation (3). Could alternative, equally scientifically justifiable covariates have been selected that would have resulted in (slightly) different conclusions being drawn?

As mentioned before, the covariates are based on the underlying changes in the 5yRx1d data. We could have worked with e.g. idealised polynomials tailored to the mean and standard deviation changes in the data. This would not affect conclusions though, since the polynomials would have to be similar to the ensemble mean/sd response to be scientifically justifiable. As we mentioned and show in the figure above in answer to the earlier comment on L146, the timeseries resulting from concatenating stationary 5-yearly GEV fits in which no constraints are put on the temporal evolution and parameters thus are truly "data driven", results in highly similar values for all three GEV parameters. This gives us confidence that our covariates are appropriate and result in satisfactory GEV-fits.

Significant changes in the conclusions would only result from GEV parameters with very different slopes than those in the current study. In our view, there is no scientifically justifiable way to impose very different parameter evolutions with time, as these would deviate strongly from the underlying data.

Line 449: Yes, I think that the framework that you have presented is very valuable and certainly the most valuable thing I got from the paper. That's the thing I am going to remember most from the paper. Given that, you may want to reframe your abstract a little?

That's a good suggestion, we have revised the abstract quite a bit and mention the framework now.

Line 529: But if you have an ensemble of several thousand members (e.g. from the weather@home model database) you should be able to determine with some statistical robustness the non-stationarity in the shape parameter right? So it is not a priori uncertain. You just need way more data to get a statistically robust estimate of the non-stationarity in the shape parameter. Are there not other papers published that show that excluding non-stationarity in the shape parameter leads to the most parsimonious structure of the GEV model for most climate related time series?

As mentioned, there is no clear indication that a non-stationary shape parameter improves the fit of the non-stationary GEV in our analysis, as we also show in the new Supplementary Sect. 2. This could be different for a different question and with a very large dataset, but in our application it seems appropriate to keep the shape parameter constant. You might be interested in Zeder et al. (2023), which focuses on effects of sample size on GEV uncertainty. The supplementary information of Zeder et al. (2023) contains much material on the role of the shape parameter.

Line 533: What if, instead of using the ensemble mean, you just used the annual mean global mean surface temperature anomaly? That would have been equally scientifically defensible and, in fact, many other papers have done exactly that. How different would your conclusions have been?

Indeed, linear scaling with GMST is a good approximation for many purposes, especially for spatially aggregated and moderate climate change settings. Record breaking and shattering is, however, very sensitive to changes in mean trend slopes, and the large amount of data we have available from the CESM2 large ensemble gives us the opportunity to add more regional and temporal flexibility to the analysis. We think it makes sense to use this regional/temporal information when we can, to increase regional specificity of the results.

Also, we hypothesise that the record-shattering precipitation results could be quite different in some regions because of the exponential relationship of extreme precipitation with (local to regional) temperature. The temperature increase and the exponent are expected to vary across locations, thus leading to different temporal evolutions of the ensemble mean 5yRx1d, that might not all be simple linear scalings of GMST – especially if local warming and the exponent are large, the 5yRx1d trend would significantly deviate from the GMST trend. Given the importance of the mean trend steepening for record shattering probability, A GMST covariate could thus have considerable impact on the effect of the mu trend on record shattering for some regions.

Lastly, by using GMST as a covariate, we might see different responses across models because the translation from GMST to regional temperature in the moisture convergence region, and from this to extreme precipitation differs across models. This would add complexity that would be hard to trace. By using precipitation data only, we know that the differences in record shattering probability are attributable to the modelled response of precipitation by this model. The question of what the effect is of model differences in how temperature translates to precipitation responses (and how these are modified by dynamics), is very interesting as well for a future paper though!

To summarise: the GMST scaling that's used in many papers is very useful for many purposes, yet, we think that the specificity we can add by using the actual variable of interest at the location of interest as our covariate, has added value for our study purpose.

Line 543: Ah good. I am glad you did that test. That's very useful to know.

:)

Line 547: And then if you really want to be ambitious, you could further expand the five fit coefficients in functions of latitude and longitude so that you can fit the GEV model to block maxima from a small domain around the cell of interest and thereby further reduce the uncertainties on the fit coefficients. For example, replace μ_0 with $\mu_{00} + \mu_{01} * d_lat + \mu_{02} * d_lon$ where d_lat is the latitude difference between the target cell and the

neighbouring cell and d_{lon} is the longitude difference between the target cell and the neighbouring cell. Assuming a linear dependence on latitude and longitude across a grid of 5x5 cells, which I think would be entirely justified, would increase the number of block maxim by a factor of 25 while increasing the number of fit coefficients only by a factor of 3 which I think would help to reduce the uncertainties in the GEV fit coefficients. I am not suggesting that you do that for this analysis. but it might be worth keeping in mind for the future.

That is definitely a good idea. Interestingly, in their attribution of the 2021 flood event in BNLG that we refer to as well, WWA chose to fit GEVs to data from multiple locations without adding a covariate to account for differences between locations (Tradowsky et al. (2023)). Although we fear that a simple linear function of lat/lon might not work for all regions – e.g. coastal grid cells might differ quite non-linearly from neighbouring non-coastal grid cells, and orography will start playing a role too –, the idea of pooling spatial data with fitting covariates to account for local differences has a lot of merit. Zeder et al. (2024) tested a GEV model over multiple grid cells where location and scale are grid cell specific, but the shape parameter varies with location.

Line 576: I like this formula towards the bottom of page 14 that describes your determination of 'record shattering'. Maybe it should be brought up into the main body of the manuscript and referred to quite early in the paper since understanding of what you mean by 'record shattering' is essential to understanding the whole paper.

Thanks, we've done that now!

SUGGESTED TYPOGRAPHICAL AND GRAMMATICAL CORRECTIONS

The suggestions below are very much just suggestions and the authors should feel free to incorporate them, or not, as they see fit.

Line 33: Replace 'degree in which' with 'degree to which'.

Thanks!

Line 42: Replace 'trends or records' with 'trends or records being broken' - the records still exist, they're just not being broken.

Thanks, we've changed it to "broken records" (L40).

Line 62: Replace 'also these will occur' with 'these will also occur'.

Done.

Line 105: Replace 'in line with' with 'consistent with'.

Yes, that's better.

Line 117: Replace 'simulated data' with 'model simulations'.

Agree.

Line 159: Replace 'are of import' with 'are important' - unless you want to sound fancy. Let's keep it fancy :) and some variation in phrases is nice too every now and then.

Line 167: Replace 'change of precipitation' with 'change in precipitation'.
Done.

Line 175: Replace 'arctic' with 'Arctic'.
We've removed the word altogether.

Line 183: Replace 'is very similar as well' with 'is also very similar'.
Agree.

Line 184: I think that it would be better to refer to these as 'zonal means' rather than 'latitudinal averages'.
Correct.

Line 200: Replace 'northern hemispheric' with 'NH'. You've defined the acronym so you might as well use it.
Good point.

Line 274: Replace 'would compound spatial extent' with 'would compound the spatial extent'.
Changed.

Line 324: Replace 'only the σ -trend is' with 'only the σ non-stationarity is'.
We prefer trend. The trend is prescribed by the covariate so that it represents the monotonous forced response. Non-stationarity is very general, and could be random fluctuations as well.

Line 326: Replace 'aspects' with 'behaviour'.
We've changed it to features.

Line 334: Replace 'the μ -trend only does not increase record shattering probability, yet, it does manifest as a lessening of the decreasing probability' with 'not only does the μ -trend increase record shattering probability, it also manifests as a reduction of the decreasing probability'. Although the phrase 'lessening/reduction of the decreasing probability' I still find confusing. You may need to give this some further thought.

Yes, it was confusingly phrased: the μ -trend only (i.e just the μ trend) does not *increase* RSP, yet, it does cause lessening of the decrease (so it increases it relative to a more steeply decreasing baseline).

We settled on this now, L379:

"the μ -trend alone is not enough to lead to a net increase in record shattering probability, yet, it does increase the probability relative to the decrease with time seen in the stationary "no trends" case."

Lien 479: Replace 'data was' with 'data were'. And likewise elsewhere. The word 'data' is plural (like the word 'values'). Its singular form is datum (value).

Done.

Lien 546: Replace 'the amount of estimated parameters' with 'the number of estimated parameters'.

Agreed.

Reviewer #3 (Remarks to the Author):

The manuscript uses extreme value statistics and CMIP6 models (with a focus on CESM2-LE) to show there is a higher probability of record shattering precipitation driven by increasing variability by the end of the century. This has implications for developing adaptation policies and mitigation strategies related to anticipating to rare events. While the focus of the manuscript is on statistics of precipitation, the authors also show physical 'realness' of their results, at very detailed levels (examples of conditions (IVT,SLP, updraft anomalies) during "record breakers" in the observational and modeled simulated results) as well as explanation of results over large geographic areas. The manuscript concludes with an analytical rationale to explain the patterns being seen. Overall, it provides compelling evidence for the importance of changing variability of precipitation, rather than just mean, which has also been highlighted in prior studies. At the same time, I have some concerns related to the strength of the results for one CMIP6 model only and the approach taken, both in the non-stationary extreme value statistics and in the comparison of the modeled to observational data, which should be further explained.

We thank the reviewer for these kind and constructive words. Below we respond to the specific individual comments.

General Comments:

One of my main concerns is that the strength of the "record shattering precipitation driven by increasing variability" is mainly true for one model only. While this conclusion is transparent in the figures and throughout the manuscript, it seems as if the importance of this should be more upfront (in the title or abstract). For example, there are some snippets of text that seem to be hidden away in the manuscript and in the supplemental section that basically say CESM2-LE data shows the most influence from scale parameter of any model, and hence variability, so the results are not robust across other future projections. BUT the conclusion holds that IF extreme precipitation variability increases strongly, record shattering probabilities respond strongly. If this is the case, there should be more specific emphasis on the single model aspect of these findings in the title and the abstract, as those are absolute in saying that shattering precipitation is driven by increasing variability in the future, across CMIP6 models.

This is a fair point. We never had the intention to hide the fact that CMIP6 models project different responses in extreme precipitation, and we do not know how much precipitation variability will change in our real climate. In addition, finding variability trends in short

observational timeseries is even harder than finding mean trends, so trying to put some observational interval on realistic variability increases is virtually impossible/has very low fidelity.

An additional difficulty here is that, statistically, low baseline variability will make the record shattering response more sensitive to changes because, in a way, the bar is lower: the level of the last record is relatively close to the mean and therefore easier to break with a smaller shift in the distribution (either mean or variability – this is what we see in NDJFM NH). That means that models with low baseline variability will most likely show less dominance of variability increases (because mean increases “work”). Models with higher baseline variability will be more “reliant” on variability increases. In addition, higher baseline variability is likely associated with higher variability increases because the highest percentiles see the strongest increases, which leads to stronger widening of already wide distributions. Hence, there is quite a lot at play here and it is outside the scope of this paper to fully disentangle all the distributional intricacies.

Nevertheless, we revise the title of the paper to reflect more explicitly that variability is a key factor (this is universally true for all models) in precipitation record shattering probabilities: **“Increasing extreme precipitation variability plays key role in future record-shattering event probability”**

In addition, we added the breakdown of probability ratio maps for the 4 non-CESM2 CMIP6 models in the supplementary information, SI Fig. 16. In a future study we hope to address the role of variability and intermodel differences therein more explicitly.

Following on, I was surprised that there was no discussion of bias correction of the GCM precipitation data. The main point of the paper is to evaluate the changing statistics of very rare events, so there needs to be a description/quantification of how the statistics of the simulated historical precipitation data preserve the statistics of the observed precipitation data over the same period of time. If the historical simulations don't reproduce the statistics of the historical observed data, then the change values across to the future can't necessarily be trusted. A qualitative comparison of the deviation around the anomalies isn't enough – the variation in the distribution – especially at the tails – needs to be quantified. Some statistical bias correction methods commonly used are linear scaling, quantile mapping, distribution mapping, and cumulative distribution function transforms. Other methods have combine approaches (developed hybrid methods) for preserving the extreme and whole ends of the distribution. Either way, if the data doesn't need to be bias corrected to retain the statistics of the distribution, this needs to be clearly shown.

We appreciate this comment and agree with the reviewer that it is absolutely essential to carefully check agreement between models and observations when claims about historical and future “real” climate change are made.

However, we do not intend to predict the future of our real climate in this study, but stay within the simulated climate realm when we analyse the characteristics of record shattering behaviour under climate change - all results from the second results subsection onwards are

contingent on the model used, and we clearly acknowledge this fact throughout the study. Moreover, when it comes to tail characteristics, bias correction is far from trivial, because the tail that has been sampled over the short observational record – for 5yRx1d we would at most have 25 values for some locations, and way fewer for most locations – is subject to large sampling uncertainty (e.g., Zeder et al., 2023). A bias correction method would aim to bring the tail of simulated extreme precipitation in close agreement with observations, but because of the large sampling uncertainty and the uneven sample sizes (n very large in the large ensembles), it is unclear whether such a correction would in fact increase the similarity of the simulated 5yRx1d distribution to the true, real-world 5yRx1d distribution.

In Figure 1 we employ observations with the purpose of showing that models do in fact produce unprecedented extremes similar in character as those we observe. We show this because it is not evident given that CMIP6 models do not resolve all processes involved in extreme precipitation, so this is meant as a justification for investigating this class of events. This figure is not intended to claim that CESM2 is comparable to observations across the board. We state more clearly now, at the end of the first results subsection in L131, that this is the case:

“Importantly, this congruence does not imply that the statistical characteristics of precipitation as simulated in CESM2-LE are comparable to observations in general: there are large model uncertainties surrounding extreme precipitation statistics and projections, and also observational uncertainty is large due to sparse coverage and short timeseries. In the following, our results thus pertain to the large ensemble climate model context in which we conduct this study. Additional work is needed to connect these findings to our real climate.”

Our analysis from there onwards focuses on the behaviour in models only. The fact that we mostly focus on probability ratios (i.e. changes relative to a baseline state) removes some of the model dependence, and the analytical rationale shows in a nondimensional way that the conceptual framework we suggest for record shattering precipitation changes holds. But, once again, we agree that a lot more rigour is needed to connect these findings to observations/our real climate.

In a paper in preparation, we employ observations and indeed use quantile mapping. We hope the reviewer will keep an eye out for that.

What validation tests did the authors do to ensure the significance and goodness of fit of their nonstationary GEV models? When fitting models with several covariates, it's best practice to ensure parsimony of the best model. This is often done with a combination of diagnostic plots, information criteria, and likelihood ratio tests. I suggest the authors test and show the significance of their model, with non-stationarity in the location and scale parameter compared to a stationary model fit as well as a comparison of a model with just nonstationarity in the scale parameter OR nonstationarity in the shape parameter to their model of nonstationary in the location and scale parameter.

Thanks for this valid comment.

In the previous version of the manuscript, Supplementary Sect. S6 showed QQplots for the GEV fits in the three regions we analyse. We see that this alone is not very rigorous as

validation, so we now added Supplementary Sect. S2, which is specifically dedicated to validation of the GEV fits to the data. For the sake of space/conciseness, we cannot show every gridcell, but we show the validation for a random selection of 20 grid cells, and also for all the grid cells in the three regions of interest.

As this supplementary section shows, the setup we use with non-stationarity in μ and σ , but not in ξ , is a very good trade-off between goodness of fit and tractability.

Specific Comments/Line-by-Line:

Related to the nonstationary GEV models – I'm also wondering why covariates related to the ensemble mean and ensemble variability were used instead of letting the form $\mu(t) = \mu_0 + \mu_1(t)$ (or some other nonlinear trend), letting the μ_1 (or σ_1) scale by the extreme data changing over time, rather than an external covariate?

We are not entirely sure we understand this comment correctly, but as we see it, our covariates do exactly that: they represent the change of the extreme data over time. See equation (4) in the revised manuscript: $g_{\mu}(t)$ is the change of the local 5yRx1d ensemble mean with time, and μ_1 is a constant that scales the "strength" of this change. Same for $g_{\sigma}(t)$, which is the change of the local 5yR1d ensemble standard deviation. So although we introduce these as covariates, they are directly derived from the data that we are fitting the distribution to.

We use covariates that prescribe the change in time, so that we can fit one coherent non-stationary GEV distribution, rather than many different ones for each timestep. In L592-600 we detailed that we do this to increase the robustness of the GEV fits, and we mention that the resulting GEV parameters are very close to what we would have obtained with fully unconstrained models (see also our answer to reviewer 2's comment about L146 and Fig. R1).

Where does CESM2-LE fit in the climate models in terms of precipitation? The authors reveal there are similar patterns to other climate models, but different projections of precipitation distribution changes. In general, what does the literature say about the variation in precipitation across the models?

A very nice comprehensive analysis on extreme precipitation biases across CMIP6 models has been done by Abdelmoaty et al. (2021), which we originally only referred to in the supplementary information. We now added this reference in the results section and the conclusions:

L234: "A comprehensive study by Abdelmoaty et al. on CMIP6 model biases in the extreme precipitation distribution showed that CESM2 and also UKESM1-0-LL are among the top 10 of the 34 models analysed in terms of similarity to precipitation observations. CESM2 and UKESM1-0-LL have significantly lower biases than ACCESS-ESM1-5 and the MPI models in mean, variability, skewness and kurtosis of extreme precipitation distributions. Particularly in the tropics where mean and variability of precipitation is generally underestimated by climate models, CESM2 and UKESM1-0-LL feature a smaller underestimation. This adds real-world relevance to the probability ratios resulting from CESM2-LE, however, it is important to keep in mind that the climate sensitivity of CESM2 is on the high end of the CMIP6 spectrum."

L502: "CESM2, the core climate model used in this study, has been shown to reproduce the distribution of extreme precipitation comparatively well. The high climate sensitivity of CESM2, however, needs to be noted when interpreting the results."

Do the patterns of shattering probabilities, relate at all to the shape parameter of the model? I'd assume there is also geographic variability of the shape parameter, usually heavier tailed around the tropics. Similarly, does the shape parameter alter the analytical results (i.e. would you see the same patterns and trends regardless of heavy, light, or bounded tail)? What is the shape parameter for the analytical results in section 3 of the paper?

This is an interesting question. The effect of the shape parameter on evolving record shattering probabilities is mostly indirect, by which we mean the following: A high shape parameter is associated with higher stationary record shattering probabilities (P_{stat}) which in turn lead to lower probability ratios because the denominator is large ($P_{\text{fut}}/P_{\text{stat}}$). Because the shape parameter is kept constant, it does not affect future shattering probabilities much – these are dependent on the changing location and scale parameter. In fact, there is a slight damping effect of a large shape parameter on record shattering, since the heavy tail corresponds to relatively low $\text{prod}_t(F(x))$ values (see eq. 2) (small chance that extreme values have not been seen before).

We added a comment to this effect in L273:

"For the sake of completeness, we comment on the role of the shape parameter ξ . Since ξ governs the tail properties of the distribution, its value affects record shattering probabilities. If ξ is constant with time, its primary effect is that increases in record shattering probability for a given μ and σ trend are damped for locations where ξ is positive and large (i.e. approaching 0.5). This behaviour is seen since a heavy tail damps the value of the last term in Eq. \eqref{eq:integralRS} -- the probability that high 5yR_{1d} values have never been seen is smaller for heavier tails. The effect, however, is small. In this study, we do not further explore the effect of ξ on record shattering."

In section 3 the shape parameter is set to 0.1, as we mention in L431 and the method section L753.

What are the limitations of the work related to resolution /modeling of tropical cyclones in the data set? How well are tropical cyclones represented in the extreme precipitation data here?

We agree that this is an important point. A study by Han et al. (2022) (figure 4) investigated tropical cyclone environments in CMIP6 models and shows that CESM2 does not do badly. The added reference to literature (see above, Abdelmoaty et al. (2021)) comparing simulated extreme precipitation distributions to observations serves to provide context on the accuracy of CESM2. This study also shows that for the tropical band, CESM2 performs comparatively well. We choose not to refer explicitly to the performance of CESM2 in reproducing tropical cyclones, since we do not explicitly consider the weather systems leading to record shattering precipitation in the different regions. Hence, we consider the

issue of resolution dependency of tropical cyclones not within the scope of our study, since we focus more on a general framework for record-shattering behaviour.

Line 606 – 607 says “it is important to view trend-separation in the GEV context in terms of location and scale” (rather than mean and standard deviation). However the section starting on line 310, is titled “The role of mean and variability for regional changes...” and the authors discuss mean and variability, and specifically call “mean trend only” (μ , dotted), and small mean trends and small variability trends, rather than using the location terminology. This happens also in lines 355 – 357 and in Section 3, analytical rationale which also discusses different mean and variability trend combinations.

This is a good point. It has to be clear that we perform the analysis in a GEV-context, and that thus location and scale are the indicators of mean and variability. We have carefully checked the manuscript and now only refer to μ and σ trends when we talk about our results. Sometimes, we do mention a mean trend (e.g. in L376: “Extreme precipitation in the NH midlatitudes (BNLG) features small mean trends with warming relative to the tropics”) in a context where we actually intend to refer to a mean trend, and are not in the GEV context. Only in captions and the conclusion section, we resort to a more general terminology so that the general gist is clear from a quick skim of the paper. We hope we filtered out all the inconsistencies now, and that the text is not confusing.

How do the authors get the confidence intervals on 500 and 5000 year events in Figure 3? They seem way too constrained to represent the uncertainty in the GEV model fit.

We use the R-package `extRemes` to estimate confidence intervals of the return levels, see manual, using the normal estimation method (as opposed to bootstrapping or likelihood profiling), which is justified with the large sample size of CESM2-LE. For the regions we show in Figure 3, we estimate the GEV parameters and return levels using 100 members, 251 years, and 8 or 9 gridcells (i.e. 200800-225900 datapoints) for each region. This means that the 1in100 or 1in1000 return levels can in fact be estimated with high confidence, leading to small confidence intervals.

Below we show schematic plots for the three regions where the different ribbons show the 95% confidence interval of the 1in100 return level (99th percentile) as a function of time for GEVs and return levels estimated from different data volumes: 1 member, 10 members and 100 members. The confidence interval based on 1 member (2008-2259 datapoints) is indeed wide, but it shrinks a lot as the amount of data is increased by a factor 100.

Fig. R4: Dependence of return level confidence intervals on ensemble size.

References

- Coles, S. (2001). Extremes of Non-stationary Sequences. In: An Introduction to Statistical Modeling of Extreme Values. Springer Series in Statistics. Springer, London. https://doi.org/10.1007/978-1-4471-3675-0_6
- Ombadi et al. (2023). A warming-induced reduction in snow fraction amplifies rainfall extremes. *Nature* 619, 305–310. <https://doi.org/10.1038/s41586-023-06092-7>
- van der Wiel & Bintanja (2021). Contribution of climatic changes in mean and variability to monthly temperature and precipitation extremes. *Commun Earth Environ* 2, 1. <https://doi.org/10.1038/s43247-020-00077-4>
- Wood (2023). Role of mean and variability change in changes in European annual and seasonal extreme precipitation events, *Earth Syst. Dynam.*, 14, 797–816, <https://doi.org/10.5194/esd-14-797-2023>.
- Zeder et al. (2023). The effect of a short observational record on the statistics of temperature extremes. *Geophysical Research Letters*, 50, e2023GL104090. <https://doi.org/10.1029/2023GL104090>
- Tradowsky et al. (2023). Attribution of the heavy rainfall events leading to severe flooding in Western Europe during July 2021. *Climatic Change* 176, 90 (2023). <https://doi.org/10.1007/s10584-023-03502-7>
- Zeder et al. (2024). Decadal to centennial extreme precipitation disaster gaps — Long-term variability and implications for extreme value modelling, *Weather and Climate Extremes*, <https://doi.org/10.1016/j.wace.2023.100636>
- Abdelmoaty et al. (2021). Biases beyond the mean in CMIP6 extreme precipitation: A global investigation. *Earth's Future*, 9, e2021EF002196. <https://doi.org/10.1029/2021EF002196>
- Han et al. (2022). Assessing the performance of 33 CMIP6 models in simulating the large-scale environmental fields of tropical cyclones. *Clim Dyn* 58, 1683–1698. <https://doi.org/10.1007/s00382-021-05986-4>

REVIEWERS' COMMENTS:

Reviewer #2 (Remarks to the Author):

Second review of "Surge in future record shattering precipitation driven by increasing variability" by Iris de Vries, Sebastian Sippel, Joel Zeder, Erich Fischer and Reto Knutti

The authors have done an excellent job in responding to my first review and the paper is now much improved and ready to be published. I think that if the review and the authors response are provided along with the paper this will further add to the quality of the manuscript as a whole. Great paper. Time for it to be published. I look forward to reading the published version.

Reviewer #3 (Remarks to the Author):

I am satisfied with the authors' detailed responses and additional analyses. I think this paper is a valuable contribution to the literature. Well done!

Dear reviewers,

Thanks very much for your positive remarks, and once again also for the helpful comments on the first manuscript!

Best,

Iris de Vries, on behalf of all authors